

# Exceptional wildfire smoke over Greece in summer 2023: a synergistic study of aerosol optical-microphysical and UVB radiative impacts

Marilena Gidarakou[1*], Alexandros Papayannis[1,2], Maria Mylonaki[3], Eleni Kralli[1], Kostas Eleftheratos[4,5], Ilias Fountoulakis[6], Olga Zografou[7], Evangelia Diapouli[7], Maria I. Gini[7], Stergios Vratolis[7], Konstantinos Granakis[7,8], Konstantinos Eleftheriadis[7], Nikolaos Evangeliou[9], Christine Groot Zwaaftink[9], Eugenia Giagka[1], Marios - Andreas Zagklis[1], Igor Veselovskii[10]

[1]Laser Remote Sensing Unit (LRSU), Department of Physics, National Technical University of Athens (NTUA), 15780 Zografou, Greece;
[2]Laboratory of Atmospheric Processes and Their Impact (LAPI), École Polytechnique Fédérale de Laussane (EPFL), 1015 Lausanne, Switzerland;
[3]Meteorological Institute, Ludwig-Maximilians-Universität München, 80539 Munich, Germany;
[4] National Kapodistrian University of Athens, Department of Geology and Geoenvironment, Athens, Greece;
[5]Biomedical Research Foundation, Academy of Athens, Athens, Greece;
[6]Research Centre for Atmospheric Physics and Climatology, Academy of Athens, Athens, Greece
[7]ERL, Institute of Nuclear & Radiological Sciences & Technology, Energy & Safety, N.S.C.R., Demokritos, Ag. Paraskevi, 15310, Greece
[8]Climate and Climatic Change Group, Section of Environmental Physics and Meteorology, Department of Physics, National and Kapodistrian University of Athens, Athens, Greece
[9]Department for Atmospheric and Climate Research (ATMOS), Stiftelsen NILU (former Norwegian Institute for Air Research), Kjeller, 2007, Norway
[10]Prokhorov General Physics Institute, Russian Academy of Sciences, Moscow, Russia

Correspondence to: Alexandros Papayannis (alexandros.papagiannis@epfl.ch), Marilena Gidarakou (marilenagidarakou@mail.ntua.gr)

**Abstract.** During summer 2023, Greece experienced one of its most severe wildfire seasons in recent decades, with widespread fires across Evros, Rodopi, Attica, the Peloponnese, and several islands. This study investigates the aerosol optical and microphysical properties, as well as the impact on ground-level ultraviolet-B (UVB) radiation over Athens, focusing on two major wildfire episodes (18–21 July and 22–25 August). A synergistic approach was deployed, combining satellite imagery (MODIS), FLEXPART simulations, ground-based remote sensing, and in situ aerosol and radiation measurements. Elevated aerosol optical depths (AOD) up to 1.2, high fine-mode fractions (>0.85), and Ångström exponents above 1.5 indicated a strong dominance of fine biomass burning aerosols. The Single scattering albedo (SSA) ranged from 0.85 to 0.98, showing enhanced absorption during biomass burning periods and weaker absorption when smoke was mixed with dust. At 320 nm, dust presence resulted in stronger absorption, with SSA below 0.8 for pure dust cases compared to smoke mixtures. Particle linear depolarization ratios (PLDR), varied between 0.03 and 0.20, with higher values (~0.10–0.20) reflecting the presence of non-spherical dust particles, and lower values (~0.03–0.08) indicating spherical smoke particles. Ground-level UVB irradiance



decreased by up to 50% during peak smoke episodes, highlighting strong aerosol radiative impacts. Concurrently, $PM_{10}$ and $PM_{2.5}$ concentrations increased to 94 and 49 µg m$^{-3}$, respectively, while organic aerosols peaked at 22.77 µg m$^{-3}$, consistent with intense fire activity. FLEXPART simulations confirmed long-range transport of smoke from active fire regions, with additional contributions from regional pollution and Saharan dust.

## 1. Introduction

Wildfires are an increasing environmental threat in the Mediterranean, driven by very high temperatures, prolonged droughts, and extreme weather events associated with global warming and climate change (Ruffault et al., 2018). These events release large amounts of aerosols and trace gases (PM, CO, $NO_x$, VOCs), affecting air quality, human health, ecosystems, and climate (Knorr et al., 2017; Mylonaki et al., 2024; Vasilakopoulou et al., 2023). Moreover, biomass burning aerosols have significant radiative effects, altering atmospheric heating rates and contributing to short- and long-term climate forcing (Intergovernmental Panel On Climate Change (IPCC), 2023).

Previous studies have documented wildfire impacts in Greece (Peloponnese 2007; Athens 2009, 2021; Evia 2021; Evros 2023) (Amiridis et al., 2012; Kaskaoutis et al., 2011, 2024; Masoom et al., 2023; Michailidis et al., 2024; Osswald et al., 2023; Turquety et al., 2009), highlighting long-range transport, elevated aerosol loadings, and complex vertical layering. For example, Kaskaoutis et al. (2011) showed long-distance transport during 2007 fires, while Amiridis et al. (2012) and Michailidis et al. (2023) reported extreme AOD and UV attenuation near urban centers. Modelling studies (Osswald et al., 2023) emphasized pollutant exceedances and public health concerns, and more recent work stressed the importance of multi-instrument observations to capture interactions between smoke, dust, and other short-lived climate forcers (Amiridis et al., 2024; Poutli et al., 2024). Nevertheless, comprehensive multi-parameter analyses of extreme smoke-dust episodes remain limited.

Beyond the Mediterranean region numerous studies have investigated biomass burning impacts worldwide. Large-scale fire events in the Amazon (Baars et al., 2012; Mayol-Bracero et al., 2002), boreal regions (Burton et al., 2012; Groß et al., 2013; Murayama et al., 2004; Ortiz-Amezcua et al., 2017; Torres et al., 2020), and Australia (Ohneiser et al., 2020; Papanikolaou et al., 2022; Sellitto et al., 2022) have demonstrated severe impacts on atmospheric composition, radiative forcing, and long-range transport of smoke plumes. For instance, the 2019-2020 Australian wildfires injected smoke into the stratosphere with measurable global radiative effects (Chang et al., 2021; Heinold et al., 2022; Sellitto et al., 2023), while Canadian boreal fires in 2023 produced trans-Atlantic smoke transport to Europe (Gidarakou et al., 2025; Reichardt et al., 2024; Veselovskii et al., 2024). Such international examples highlight that biomass burning is not a regional problem but a global phenomenon with far-reaching implications. In this context, our study complements previous work by providing a detailed analysis of extreme wildfire-dust interactions in the Eastern Mediterranean, a climate-change hotspot.

In this paper we focus on summer of 2023 which was one of the most catastrophic wildfire seasons in Greece. The Evros wildfire, ignited on 19 August, burned over 90,000 hectares under extreme heat (>40 °C) and strong winds, making it the



largest single wildfire recorded in the European Union since 2000 (EFFIS). Beyond its direct destruction, this event highlights the increasing likelihood of extreme wildfires under a warming Mediterranean climate and the need to understand their atmospheric consequences. Furthermore, we note that during summer seasons our study area is affected by wildfires smoke particles from the Balkan region (Papayannis et al., 2014).

A notable feature of the 2023 wildfires season was the simultaneous presence of intense wildfire smoke aerosols and Saharan

dust particles, as revealed from the analysis of MODIS data (not shown here). Thus, smoke plumes from Evros reached central and southern Greece, including the Athens basin, while dust outbreaks contributed coarse particles, creating complex mixtures of fine-mode smoke (r < 0.5 μm) and coarse mineral dust (Dubovik and King, 2000; Holben et al., 1998). Such episodes are particularly relevant due to their distinct optical, microphysical, which influence air quality and climate forcing (Liu et al., 2022, 2020; Masoom et al., 2023). Beyond their atmospheric effects, they also pose risks for human health and ecosystems:

exposure to fine particles from wildfire smoke has been linked to respiratory and cardiovascular diseases (Chirizzi et al., 2017; Mylonaki et al., 2024), while dust-smoke mixtures may worsen health impacts and reduce agricultural productivity (Singh et al., 2022).

This study focuses on transport of wildfire smoke and Saharan dust particles over Athens during 17 July - 30 August 2023, emphasizing on the Evros wildfires event. Using a synergistic approach that combines active and passive remote sensing, in

situ air quality and solar UVB measurements, satellite observations, and FLEXPART modelling, we provide a detailed characterization of aerosol vertical distribution, optical and microphysical properties, and radiative impacts. This work extends previous research on Greek wildfire episodes and highlights the atmospheric consequences of increasingly catastrophic events in a warming climate.

The paper is structured as follows: Section 2 describes the instrumentation and methodology, including lidar systems, sun-sky-

lunar photometric retrievals, air quality monitoring, solar irradiance measurements, satellite observations and FLEXPART simulations. Section 3 presents the results, covering event evolution, aerosol optical and microphysical properties, and their impact on both columnar and near-surface conditions. Section 4 summarizes the main conclusions of the study.

## 2. Methodology and Instrumentation

The presented data were collected during the summer campaign of the European Partnership on Metrology project

BIOSPHERE (Metrology for Earth Biosphere: Cosmic rays, UV radiation and fragility of ozone shield) (Pierrard et al., 2025) which aims to develop and use novel instrumentation and methods to assess how the increasing ionization of the atmosphere affects the human and ecological health on our planet.

The main observation site was the National Technical University of Athens (NTUA) (37.96º N, 23.78º E; ~ 212 m a.s.l.), where two lidar systems, a Raman lidar and a depolarization lidar, along with a CIMEL sun-sky-lunar photometer were operated.

Additionally, aerosol and air quality measurements were obtained at the Demokritos station (DEM) (37.99º N, 23.81º E ~ 270 m a.s.l.), which provided data from an aethalometer, a Time-of-Flight Aerosol Chemical Speciation Monitor (ToF-ACSM,



Aerodyne Research Inc., USA) for monitoring aerosol chemical composition, as well as 1-h data of $PM_{10}$ and $PM_{2.5}$ concentrations. At the Biomedical Research Foundation of the Academy of Athens (BRFAA) (37.99º N, 23.78º E; ~ 180 m a.s.l.), a Brewer spectrophotometer contributed measurements of UVB radiation and aerosol optical depth. To support the
analysis of wildfire impacts and atmospheric transport, FLEXPART dispersion modelling and wildfire detections from MODIS (FIRMS) satellite imagery data were also used.

### 2.1 Lidar systems

Two lidar systems were used to retrieve aerosol optical properties: the DEPOLarization lidar systEm (DEPOLE) and the elastic-Raman lidar system aErosol and Ozone Lidar system (EOLE). DEPOLE is based on a pulsed Nd: YAG laser emitting
at 355 and 532 nm with linear polarization purity more than 99.5%, achieved using a polarizing filter. The elastically backscattered lidar signals are collected at both wavelengths by a 200 mm diameter Dall-Kirkham (f/4) Cassegrainian telescope and are separated into parallel and cross-polarization components using polarizing beam splitter cubes. DEPOLE achieves full overlap at approximately 500 m a.s.l. (Papayannis et al., 2020). DEPOLE provides vertical profiles of the aerosol elastic backscatter coefficient and volume and particle linear depolarization ratios VLDR and PLDR, respectively, at 355 and 532
nm, as well as the aerosol Ångström exponent between these two wavelengths ($AE_{355/532}$).

The EOLE lidar system is based on a pulsed Nd: YAG laser emitting simultaneously at 355, 532, and 1064 nm. The receiving unit includes a 300 mm diameter (f/2) Cassegrainian telescope, which collects both the elastically backscattered signals and the vibrational-rotational Raman signals generated by atmospheric $N_2$ at 387 and 607 nm, and $H_2O$ at 407 nm. EOLE provides vertical profiles of the aerosol backscatter coefficients at 355, 532, and 1064 nm, and extinction coefficients at 355 and 532
nm, as well as the water vapor mixing ratio in the troposphere. Additionally, EOLE provides vertical profiles of intensive aerosol parameters, including the lidar ratio (LR) at 355 and 532 nm, and aerosol Ångström exponents derived from extinction ($AE_{e355/532}$) and backscatter ($AE_{b355/532}$, $AE_{b532/1064}$) coefficients. The system reaches full overlap at approximately 800 m a.s.l.. Based on these observations and using well-known methodologies, we can separate the lidar signals between aerosols and clouds, spherical and non-spherical particles in mixed aerosol layers (Ansmann et al., 2012, 2019; Tesche et al., 2009).

### 2.2 CIMEL sun-sky-lunar photometer data

Sun-sky-lunar photometric measurements are continuously conducted at NTUA, using a CE318-T sun-sky-lunar photometer, part of the Aerosol Robotic Network (AERONET) and operated by the Laser Remote Sensing Unit (LRSU). The instrument performs automated observations of direct solar irradiance and sky radiance in both the almucantar and principal plane, typically every 15 and 30 minutes, respectively, across multiple spectral channels (including 340, 380, 440, 500, 670, 870,
940, and 1020 nm). The instrument performs also nighttime measurements using lunar irradiance. In this study, we used Level 2.0 (cloud-screened and quality-assured) AERONET Version 3 retrievals, including aerosol optical depth (AOD), Ångström exponent (AE), single-scattering albedo (SSA), fine/coarse AOD, and aerosol volume size distributions (VSD), to characterize the optical and microphysical columnar properties of aerosols over Athens. Lunar AOD data were additionally used to extend



the temporal coverage during night periods with elevated aerosol presence. For days with extreme smoke loads (e.g., 24 and
27 August 2023), Level 1.0 data were used for direct sun products due to the automatic cloud-screening algorithm discarding
valid data as cloudy under the high temporal variability of wildfire plumes. Correspondingly, Level 1.5 inversion products
(SSA, VSD) were used on these days, while Level 2.0 products were used otherwise.

## 2.3 Air Quality data

In addition to column-integrated optical and microphysical properties, in situ aerosol measurements were also used to better
characterize the smoke plume's chemical and optical properties. The aerosol absorption coefficient was obtained using an
aethalometer (AE33, Magee Scientific, USA), operating at seven wavelengths (370, 470, 520, 590, 660, 880, and 950 nm).
The mass concentration of equivalent black carbon (eBC) was calculated based on the Mass Absorption Coefficient (MAC)
values provided by the instrument manufacturer. The AE33 sampled through a $PM_{10}$ inlet at 880 nm with a 1-minute resolution,
and the obtained measurements were subsequently averaged to hourly values (Diapouli et al., 2017; Stathopoulos et al., 2021).
The measured eBC was divided into two fractions (Biomass Burning and Fossil Fuel) using the model proposed by Sandradewi
et al. (2008). Furthermore, the non-refractory chemical composition of fine aerosols, including organics, sulfate ($SO_4^{2-}$), nitrate
($NO_3^-$), ammonium ($NH_4^+$), and chloride ($Cl^-$) was monitored using a time-of-flight aerosol chemical speciation monitor (ToF-
ACSM, Aerodyne Research Inc., USA) with a time resolution of 15 minutes (Fröhlich et al., 2015; Zografou et al., 2022).
$PM_{2.5}$ and $PM_{10}$ hourly mass concentrations were also obtained by the use of a Laser Aerosol Spectrometer (LAS) 3340A (TSI
Inc., USA), coupled with gravimetric analysis of 24-h $PM_{10}$ and $PM_{2.5}$ samples collected on Teflon filters by the use of
reference low volume samplers ($PM_{10/2.5}$ SEQ 47/50-CD with Peltier cooler, Sven Leckel GmbH, Germany), according to
EN12341. LAS provides the aerosol number concentration at different sizes, up to 10 μm. Hourly averages of the number
concentrations were converted to volume concentrations. $PM_{2.5}$ and $PM_{10}$ mass concentrations on an hourly basis were then
calculated by integrating the volume concentrations over the respective size range and converting to mass through comparison
with the 24-h average $PM_{10}$ and $PM_{2.5}$ concentrations obtained gravimetrically.

## 2.4 Brewer spectrophotometer data

In this study, we analysed UV radiation measurements (315 - 325 nm) for the period 16/7-30/8/2023 obtained by a Brewer
MKIV spectrophotometer operating on a daily basis at the Biomedical Research Foundation of the Academy of Athens
(BRFAA; 37.99º N, 23.78º E; 180 m a.s.l.). The institute is located in a green area, approximately 4 km away from the city
center (Eleftheratos et al., 2021) and the NTUA site. The instrument is calibrated on a biennial basis by International Ozone
Services Inc., Toronto, Ontario, Canada (www.io3.ca; last access July 12, 2025). The last two calibrations took place in Athens,
Greece in 2023 and in El Arenosillo, Spain in 2025. Additionally, every two months, UV spectra from the Brewer are calibrated
using a set of three 200 W lamps that are traceable to the National Institute of Standards and Technology
(https://www.nist.gov/). The spectra used in this study have been quality controlled and assured as described in Masoom et al.,
(2023) (Masoom et al., 2023).





The 315 – 325 nm wavelength range has been chosen because at these wavelengths, for relatively high aerosol loads (e.g., during wildfire or dust events), the effect of aerosols is dominant over the effect of ozone. For shorter wavelengths (<315 nm) variations and uncertainties in total ozone induce higher uncertainties regarding the quantification of the impact of aerosols. We consider that the impact of aerosols at these wavelengths is representative for their impact on UV-B (290 – 315 nm integral). Hereinafter, the 320 nm irradiance is referred to as UV-B irradiance.

The AOD data from the Brewer have been retrieved at selected wavelengths used to measure columnar ozone (306.3 nm, 310.1 nm, 313.5 nm, 316.8 nm, 320.1 nm) according to the methodology described in López-Solano et al. (2018) (López-Solano et al., 2018). Comparison with the AOD at 340 nm from the CIMEL (extrapolated to the desired wavelengths using the 340 - 440 nm AE) for the period 2023 - 2025 revealed an average agreement better than 0.05 with a standard deviation ranging between ~ 0.05 (for 320. 1 nm) and ~ 0.1 (for 306.3 nm) (Figure S1).

### 2.5 Satellite data

Active fire data during the study period were obtained from the Moderate Resolution Imaging Spectroradiometer (MODIS) instruments aboard the Terra and Aqua satellites (Giglio et al., 2016). These data were accessed via the Fire Information for Resource Management System (FIRMS) platform (*https://firms.modaps.eosdis.nasa.gov/map*). To ensure high confidence in the identification of active fire locations, only detections with a confidence level greater than 80% were considered. This threshold was chosen to identify regions affected by biomass-burning emissions and to confirm that the air masses reaching Athens passed over fire-affected areas enriched with smoke aerosols (Justice et al., 2002; Vadrevu and Lasko, 2018).

### 2.6 Modelling and algorithms

#### 2.6.1 Microphysical Aerosol Properties

Aerosol microphysical properties within dust and/or biomass-burning layers were retrieved using the regularization inversion technique (Müller et al., 1999), based on lidar profiles of extinction (355-532 nm), backscatter (355, 532, 1064 nm), and depolarization (532 nm). The inversion provided the effective radius ($R_{eff}$), number concentration (N), surface area (S), volume (V), and the real (mR) and imaginary (mI) parts of the refractive index. The refractive index was assumed constant across the 355-1064 nm range and averaged over particle size. Particles were considered as spheres (PLDR < 10%) or spheroids (PLDR > 10%). Uncertainties are estimated at ±0.05 for mR, ±50% for mI and N, and below 20% for Reff, S, and V (Müller et al., 2005).

#### 2.6.2 Spectral surface solar radiation

To quantify the impact of aerosols on UV-B irradiance, a comparison between measured and modelled irradiances has been performed. Since the irradiance at wavelengths that are shorter than ~ 315 nm is affected strongly by variations in total ozone (and the corresponding uncertainties in the used total ozone values), we focused on the irradiance at 320 nm (315 nm - 325 nm





average). Radiative transfer simulations were performed using the disort (DIScrete Ordinate Radiative Transfer) pseudo-spherical approximation (Buras et al., 2011) of the UVSPEC radiative transfer model that is included in the libRadtran v2.0.6 package (Emde et al., 2016). The simulations were performed for the coordinates of the BRFAA, for the exact time of the measurements, using the columnar ozone measured by the Brewer as input. Two sets of simulations were performed: (i) simulations assuming cloud-free and aerosol-free skies, and (ii) simulations assuming cloud-free skies, for the measured AOD at 320 nm, and for various SSA values (0.6 - 1 with a step of 0.02). The former simulations were compared to the measured irradiances to quantify the overall aerosol effect. The latter were also compared to the measured irradiances, this time to estimate the SSA at 320 nm as described in (Bais et al., 2005).

Climatological profiles of atmospheric molecules corresponding to mid-latitude summer (Anderson et al., 1986) and the extraterrestrial spectrum proposed by (Kurucz, 1994) were used for all simulations. More detailed discussion for the UV-B modelling and the conditions that ensure the comparability between the measured and modelled irradiances can be found in Masoom et al. (2023).

### 2.6.3 FLEXPART modelling

To identify aerosol sources in the case studies (Section 3), we used the FLEXPART v10.4 Lagrangian model (Pisso et al., 2019; Stohl et al., 1998), driven by ECMWF ERA5 data (Hersbach et al., 2020) (0.5° resolution, hourly, 137 vertical levels). Backward simulations ("retroplume" mode) released particles from 0-4000 m at NTUA and tracked them 30 days back, covering typical lifetimes of BC and dust (~1 week). FLEXPART computed source-receptor matrices (footprints), which, when combined with emission inventories, indicate the contribution of each source region. The model accounts for gravitational settling, dry/wet deposition, turbulence, and deep convection, offering a more complete representation than simple trajectories. Anthropogenic BC sources were derived from the ECLIPSEv6 inventory (Klimont et al., 2017) (e.g., transport, industry, domestic combustion), while biomass burning used GFEDv4 (Giglio et al., 2013). Dust emissions were estimated with FLEXDUST (Groot Zwaaftink et al., 2022), using ECMWF data at 0.25° resolution and including particles from 0.2 to 20 μm. FLEXPART has been widely applied for BC and dust transport modelling and source appointment in various regions, in particular over Europe (Evangeliou et al., 2021; Gidarakou et al., 2024; Groot Zwaaftink et al., 2022).

### 2.6.4 Aerosol Mass Concentration

Aerosol mass concentration profiles for dust and non-dust components were estimated using the POLIPHON algorithm (Ansmann et al., 2012; Mamouri and Ansmann, 2014; Tesche et al., 2009), which combines depolarization lidar and sun photometer data. Depolarization lidar distinguishes dust from non-dust aerosols from aerosol backscatter and depolarization profiles, while sun photometry provides fine/coarse-mode AODs and microphysical properties (volume and surface area). Mass concentrations were calculated using aerosol-specific properties like PLDR, lidar ratio, density, and volume-to-AOD ratios. To this end we used a coarse-mode density of 2.6 g cm$^{-3}$ (dust) (Hess et al., 1998; Proestakis et al., 2024) and a fine-mode density of 1.35 g/cm$^{-3}$ (smoke) (Engelhart et al., 2012; Reid et al., 2005). The method remains effective even when lidar





and photometer data are not strictly collocated, especially for stable dust events. Total uncertainty in retrieved mass concentrations is ~36-40% and assumed mass densities (±20%). Volume-to-AOD ratios may vary by up to 10% for dust and

20% for smoke (Ansmann et al., 2012, 2019).

## 3. Results

### 3.1 Description of the wildfire/dust events (17 July - 31 August 2023)

Between 17 July and 31 August 2023, Greece experienced one of its most severe wildfire seasons in recent decades, with extensive fires affecting both the mainland and several islands. Major wildfire activity was recorded in regions such as Evros,

Rodopi, Attica, Central Greece, the Peloponnese, and islands like Rhodes, Corfu, and Evia (Figure 1a). According to the European Forest Fire Information System (EFFIS), approximately 170.000 hectares were burned during this period. The most catastrophic event occurred in Evros region, northeastern Greece, where a wildfire that began on 19 August was further intensified by strong northeasterly winds and ultimately burned over 93.000 hectares, surpassing the previous national record, and becoming the largest wildfire ever recorded in the European Union since 2000. The Copernicus Atmosphere Monitoring

Service (CAMS) reported that Greece's wildfire emissions in July 2023 reached record-breaking levels, with over 1 megaton of carbon released by 25 July, nearly twice the previous record from 2007. These emissions coincided with an intense heatwave, characterized by temperatures exceeding 40 °C and prolonged drought, which created ideal conditions for wildfire ignition and spread.

On 18 July, the Moderate Resolution Imaging Spectroradiometer (MODIS) aboard NASA's Aqua satellite captured a true-

color image of active burning fires near Athens, with a thick gray smoke plume being transported southwestward by strong winds (Figure 1b). This event marked the onset of significant biomass-burning influence over Athens.



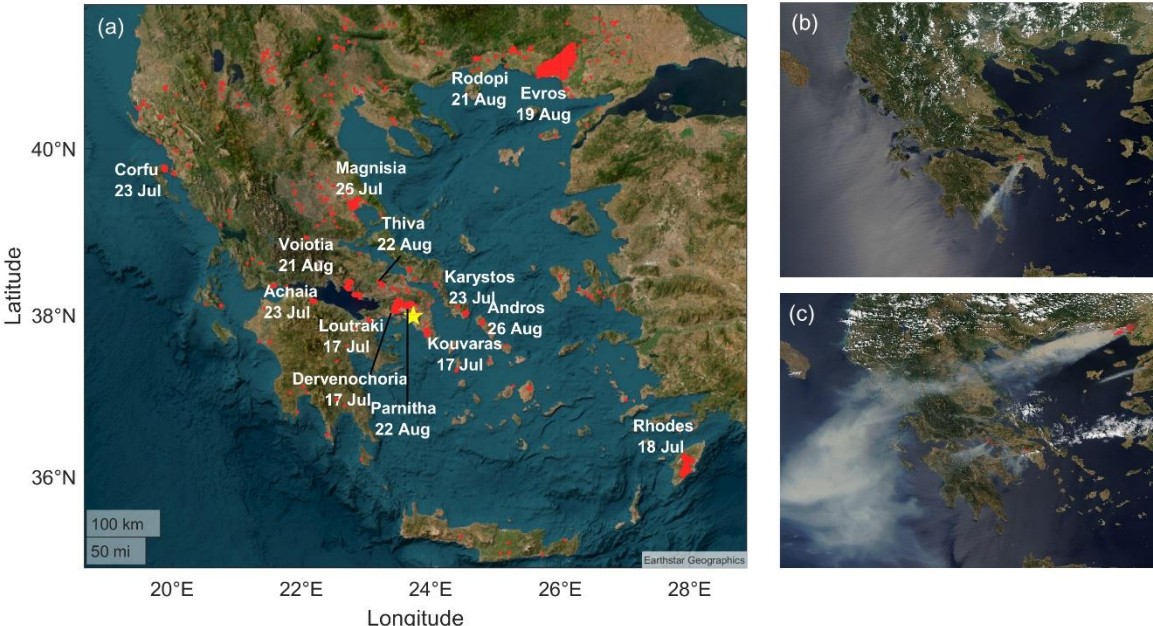

**Figure 1.** (a) Map of Greece illustrating wildfires detected by Aqua MODIS NASA Satellite imagery from 17 July to 31 August 2023 and satellite image showing wildfires over Greece (b) afternoon 18 July 2023 and (c) afternoon 22 August 2023 - data from Aqua MODIS, NASA.

Smoke plumes from these fires were transported across the Aegean Sea and Athens, reaching altitudes up to ~6 km. The elevated aerosol layers observed over Athens during this period resulted from both fine-mode biomass burning (BB) particles and coarse-mode Saharan dust intrusions.

FLEXPART simulations and MODIS fire detection data provided clear evidence of the origin and evolution of these intrusions (Figure S2, S3). Between 20 and 25 July, air masses arriving over Athens originated from North Africa, including Morocco, Tunisia, and northern Algeria, merging over the central Mediterranean before reaching Greece. Consequently, aerosols over Athens during this period were a mixture of smoke, marine aerosols, and mineral dust, primarily below 6 km height.

From 21 to 29 August, the aerosol load over Athens was dominated by intense smoke conditions with minimal dust influence. More precisely on 22 August, MODIS imagery once again revealed a dense smoke plume originating from large fires in Northern Greece, visibly extending southward toward Athens, emphasizing the persistence of biomass-burning influence (Figure 1c).

These observations were supported by multi-instrument aerosol characterization, including multiwavelength Raman and depolarization lidars, sun-sky-lunar photometer measurements, and in situ observations of black carbon (BC), elemental carbon (EC), $PM_{10}$, $PM_{2.5}$, aerosol chemical composition, and UVB radiation data.

Synoptic meteorological conditions marked by extreme heat, low relative humidity, and persistent northeasterly winds, not only facilitated the spread of fires but also supported the transport of smoke and dust. In some cases, pyrocumulonimbus clouds




further enhanced vertical and horizontal dispersion of aerosols. The combined influence of wildfire smoke and Saharan dust led to exceptionally high aerosol loads over Athens, with significant implications for air quality, and public health. These events highlight the critical need for continuous, multi-instrument aerosol monitoring to assess the impacts of extreme
atmospheric events.

In summary, the combined influence of biomass-burning aerosols and Saharan dust led to exceptionally high aerosol loads over Athens during several episodes in July and August 2023, with serious implications for air quality, radiative forcing, and public health (Mylonaki et al., 2024). These extreme events underscore the critical need for continuous, multi-instrument atmospheric monitoring to assess the evolving impacts of climate-driven hazards on regional air quality.

**3.2 Impact on air quality and columnar aerosol properties**

Figure 2 presents the temporal evolution of aerosol optical properties (Fig. 2a-b) and chemical composition (Fig. 2c-e) over Athens from 17 July to 31 August, encompassing a period of intense atmospheric perturbations related to regional wildfires and long-range dust transport from Sahara. Figure 2a presents the AOD from the CIMEL sun-sky-lunar photometer (1020-340 nm range), alongside the AE calculated from the 440/870 nm pair. Throughout the observation period, AOD values reveal
moderate to high aerosol loads, with pronounced peaks occurring in late August. AE values mostly remain above 1, indicating a dominant contribution from fine-mode particles, particularly during the high-AOD events observed between 21 and 30 August, during the wildfires that took place in the Evros region. The combination of elevated AOD at shorter wavelengths and enhanced AE suggests the presence of absorbing, fine-mode aerosols, likely associated with smoke from the widespread wildfires that affected mainland Greece and surrounding regions during this time. These findings align with previous
observations of increased fine-mode AOD and AE during summer wildfire outbreaks in Greece (Kazadzis et al., 2007; Masoom et al., 2023; Michailidis et al., 2024; Raptis et al., 2020).



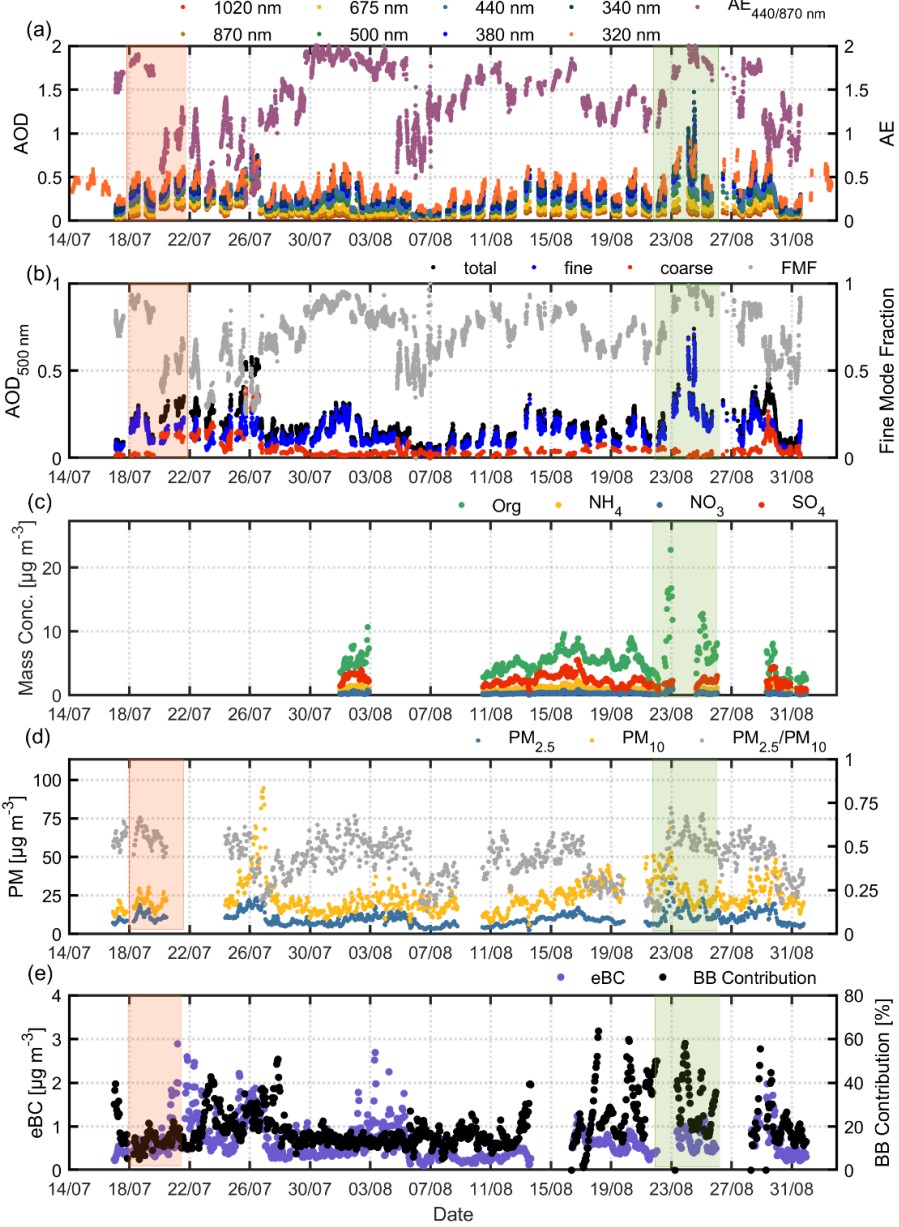

**Figure 2.** Time−series of hourly averaged values of (a) AOD for various wavelengths (340 - 1020 nm) and AE at 440/870 nm obtained by the CIMEL sun-sky-lunar photometer and AOD at 320 nm obtained by the Brewer spectrophotometer (b) AOD and AE at 500 nm along with the Fine Mode Fraction (FMF) obtained by the CIMEL sun-sky-lunar photometer (c) mass concentration of various chemical composition (Org, $NH_4^+$, Cl, $NO_3^-$) obtained by ToF - ACSM (d) 1-h PM2.5 and PM10 concentration (e) eBC concentration and BB contribution obtained by the aethalometer.



Figure 2a presents the AOD from the CIMEL sun-sky-lunar photometer (1020-340 nm range), alongside the AE calculated
from the 440/870 nm pair. Throughout the observation period, AOD values reveal moderate to high aerosol loads, with
pronounced peaks occurring in late August. AE values mostly remain above 1, indicating a dominant contribution from fine-
mode particles, particularly during the high-AOD events observed between 21 and 30 August, during the wildfires that took
place in the Evros region. The combination of elevated AOD at shorter wavelengths and enhanced AE suggests the presence

of absorbing, fine-mode aerosols, likely associated with smoke from the widespread wildfires that affected mainland Greece
and surrounding regions during this time. These findings align with previous observations of increased fine-mode AOD and
AE during summer wildfire outbreaks in Greece (Kazadzis et al., 2007; Masoom et al., 2023; Michailidis et al., 2024; Raptis
et al., 2020).

Figure 2b presents the temporal evolution of total, fine-mode, and coarse-mode AOD at 500 nm, alongside the Fine Mode

Fraction (FMF), which quantifies the dominance of submicron aerosol particles. The FMF varied substantially throughout the
study period (17 July-31 August), reflecting the changing nature and sources of aerosols over Athens. From 17 to 19 July,
FMF ranged between 0.69 and 0.93, suggesting the predominance of fine-mode particles, potentially influenced by regional
pollution or transported smoke. A notable shift occurred between 20 and 26 July, when FMF dropped significantly (0.37-0.73),
reaching values as low as ~0.2-0.4 on 23 and 25 July. This indicates a stronger contribution from coarse-mode aerosols,

consistent with dust intrusion events, which are frequent over Greece during summer (Masoom et al., 2023).

In the period from 26 July to 5 August, the FMF increased again to values around 0.75, indicating the fine- mode dominance
once again. A brief decline on 6 August (FMF < 0.5) was followed by a more stable period from 7 to 22 August, where FMF
fluctuated between 0.47 and 0.96, with a mean around 0.7. This pattern likely reflects a mix of background aerosols and
contributions from regional BB. A pronounced shift occurred between 23 and 26 August, when FMF exceeded 0.9, peaking

on 24 August. This peak coincides with the major wildfire event in the Evros region, one of the most intense fire episodes of
the summer in Greece, strongly impacting the aerosol load over Athens with freshly emitted fine particles.

The consistently high FMF values during the latter part of August, together with the concurrent rise in fine and total AOD,
indicate a substantial contribution from wildfire smoke. Similar aerosol signatures have been reported in earlier fire-affected
episodes in Greece and the eastern Mediterranean, where increased FMF and AE values are linked to enhanced black carbon,

organic aerosols, and secondary fine particles (Kaskaoutis et al., 2021; Liakakou et al., 2020; Paraskevopoulou et al., 2014).
In contrast, the relatively limited coarse-mode AOD throughout most of August suggests a negligible impact from Saharan
dust during this period.

Figure 2c presents the mass concentrations of selected non-refractory $PM_1$ chemical species, organics (Org), ammonium
($NH_4^+$), sulfate ($SO_4^{2-}$), and nitrate ($NO_3^-$) measured by the ToF-ACSM during August 2023. Organic aerosols clearly dominate

the chemical composition, with a mean concentration of 5.45 µg m$^{-3}$, ranging from 1.32 to 22.77 µg m$^{-3}$. The highest organic
levels were recorded between 22 and 26 August, peaking on 22 August, in accordance with the intense wildfires in northeastern
Greece. These elevated concentrations highlighting the major influence of biomass burning - fine-mode aerosol loading over
Athens during this period. $SO_4^{2-}$ displayed a more stable background behaviour, with a mean of 2.23 µg m$^{-3}$ (range: 0.50-5.56



μg m$^{-3}$), showing moderate increases during selected episodes, likely linked to regional transport and secondary aerosol

formation processes. On the other hand, NO$_3^-$ concentrations remained relatively low and variable, averaging 0.89 μg m$^{-3}$ (range: 0.16-2.34 μg m$^{-3}$), consistent with its volatility and the typically high temperatures in Athens during summer that limit its partitioning to the particle phase (Zografou et al., 2022). NH$_4^+$ also exhibited low concentrations (mean: 0.33 μg m$^{-3}$, range: 0.09-2.22 μg m$^{-3}$) and followed similar temporal trends to SO$_4^{2-}$, supporting the likely presence of ammonium sulfate or ammonium nitrate components during secondary formation episodes. These observations highlight the dominant role of

organic aerosols during fire events, while secondary inorganic species such as SO$_4^{2-}$ and NH$_4^+$ contributed more steadily to the PM$_1$ composition throughout August.

Figure 2d presents the hourly mean concentrations of PM$_{2.5}$ and PM$_{10}$, obtained from real-time optical measurements together with gravimetric analysis of PM filter samples during July and August 2023. The average concentrations were 10.37 μg m$^{-3}$ for PM$_{2.5}$ and 22.76 μg m$^{-3}$ for PM$_{10}$, with values ranging from 2.50 to 49.41 μg m$^{-3}$ and 6.07 to 94.60 μg m$^{-3}$, respectively.

Elevated PM$_{2.5}$ concentrations were evident during 18-22 July (orange shading) and 22-26 August (green shading), coinciding with periods of enhanced AOD (Figure 2a,b) and changes in aerosol size/composition, thus, reflecting significant aerosol loading events. During 18-22 July, the PM$_{2.5}$/PM$_{10}$ ratio remained around 0.5, suggesting a mixture of fine smoke particles and coarse dust. In late August, the ratio increased to 0.6-0.7, peaking at 0.7 on 22 August, consistent with the dominance of biomass burning particles. The concurrent increase of PM$_{10}$ concentrations also points to contributions from coarse particles,

likely resuspended dust, or aged smoke plumes.

To further quantify the relationship between particulate matter and columnar aerosol loading, we computed the Spearman correlation between hourly PM$_{2.5}$ and PM$_{10}$ concentrations and AOD at 500 nm (Figure S4). PM$_{2.5}$ shows a moderate positive correlation with AOD ($\rho = 0.60$, $p \ll 0.01$), consistent with the strong influence of fine smoke particles, while PM$_{10}$ exhibits a weaker but still significant correlation ($\rho = 0.20$, $p \ll 0.01$), reflecting contributions from coarse particles such as dust. These

results confirm that surface PM enhancements generally coincide with periods of elevated aerosol loading during the wildfire and dust events. Enhanced eBC concentrations (~1.0 μg m$^{-3}$ on 21-24 August; Figure 2e), together with the sharp increase in the BB fraction (>50%), further confirm the strong wildfire influence. Outside the wildfire periods, lower BB% values suggest mixed traffic and background sources.

### 3.3. Transport of fresh smoke and dust (the case of 18-21 July 2023)

FLEXPART footprint emission sensitivities for BC from18 to 21 July 2023 (12:00-18:00 UTC) indicate that air masses arriving over Athens between 0.5 and 4.0 km altitude were strongly influenced by biomass burning (wildfires) activity, particularly originating from southern and northeastern Greece. The footprint maps (Figure 3a-d) show increased residence time over active wildfire regions such as Dervenochoria, Greece (Figure 1b), supporting the transport of smoke plumes from local forest fires. Additional emission sensitivities over the Balkans and Central Europe suggest potential contributions from

regional air pollution sources.

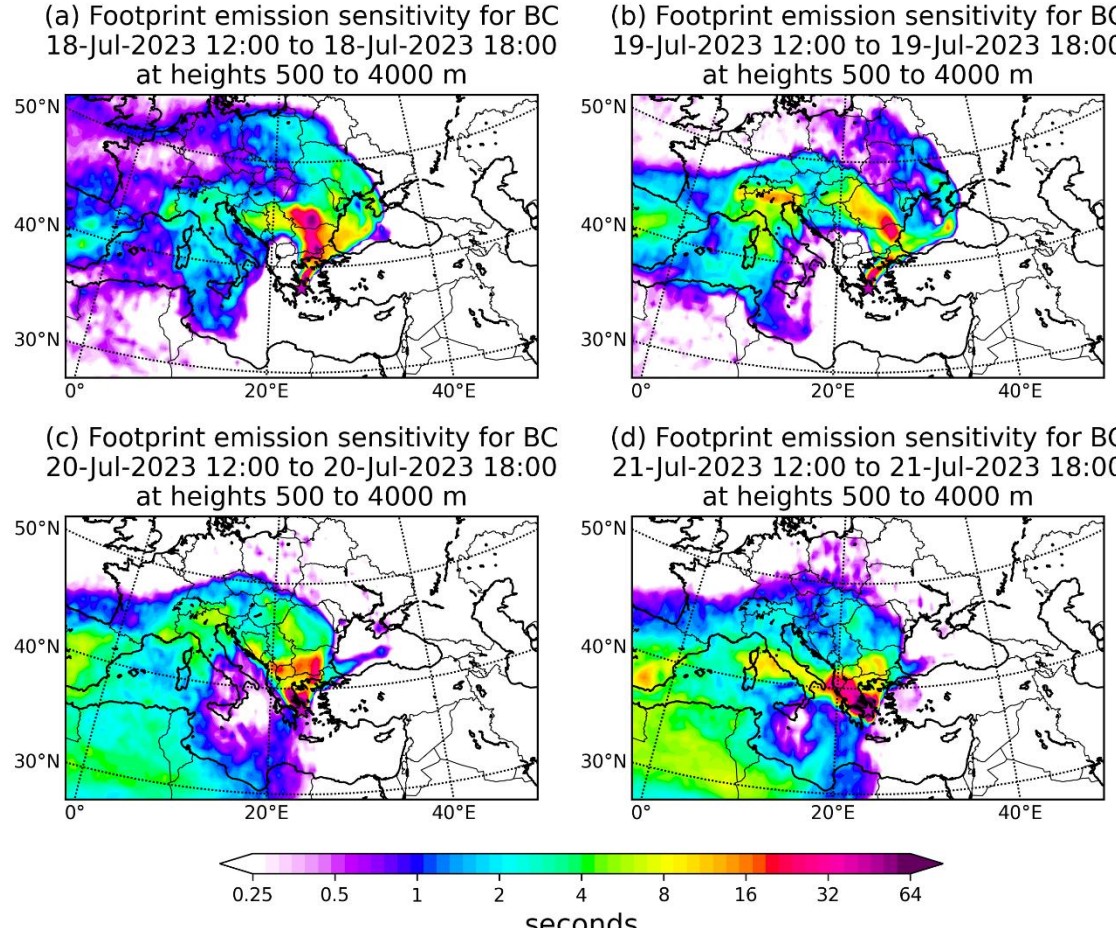

**Figure 3.** Footprint emission sensitivities for BC calculated using FLEXPART for air masses arriving in Athens between 0.5 and 4.0 km from 18 to 21 July 2023 (12:00 UTC to 18:00 UTC).

From 20 July onward, the footprint emission sensitivities extend over North Africa, indicating the influence of Saharan dust during the latter part of this period. This shift in source influence from predominant local biomass burning to a combination with mineral dust corresponds with elevated $PM_{2.5}$ concentrations, increased $PM_{2.5}/PM_{10}$ ratios (~0.6) and enhanced organic aerosol fractions observed during the episode.

Figure 4 presents the spatio-temporal evolution of the range-corrected lidar signal (RCS, in arbitrary units) at 1064 nm, obtained from the EOLE system between 18 and 21 July 2023, from ~0.5 to 6.0 km height a.s.l. On 18 July, two distinct aerosol layers were observed above the Planetary Boundary Layer (PBL): one extending from 0.5 to 2.0 km and another between 2.5 and 3.0 km height, with a very thin, detached layer around ~3.8 km height during daytime hours, likely related to transported smoke. During nighttime, the lower aerosol layer persisted, while two thin layers around 2.5 km and 3.8 km remained visible, maintaining a similar vertical structure.



On 19 July, a more stable situation was observed with a dominant aerosol layer from ~0.5 to 2.0 km height and a thin, elevated
one, between 2.0 and 3.0 km. As the day progressed into the evening, the elevated layer appeared to descend slightly. During
nighttime, the PBL height remained, while a faint layer was observed aloft, consistent with aged smoke residues.

A more dynamic picture emerged on 20 July, coinciding with the arrival of dust particles. The lidar signals showed increased
atmospheric mixing and vertical redistribution of aerosols. Early in the day, several distinct layers were identified: from 0.5 to
1.5 km, 1.5 to 3.0 km, 3.0 to 3.5 km heights, and thinner layers between 3.5 and 5.0 km. As the day progressed, these thin
upper-tropospheric layers descended and merged with lower ones, forming more compact and optically dense features.
Between 19:00 and 21:40 UTC, cloud formation was observed near 2.0 km height, likely induced by the interaction of biomass
burning particles and mineral dust particles.

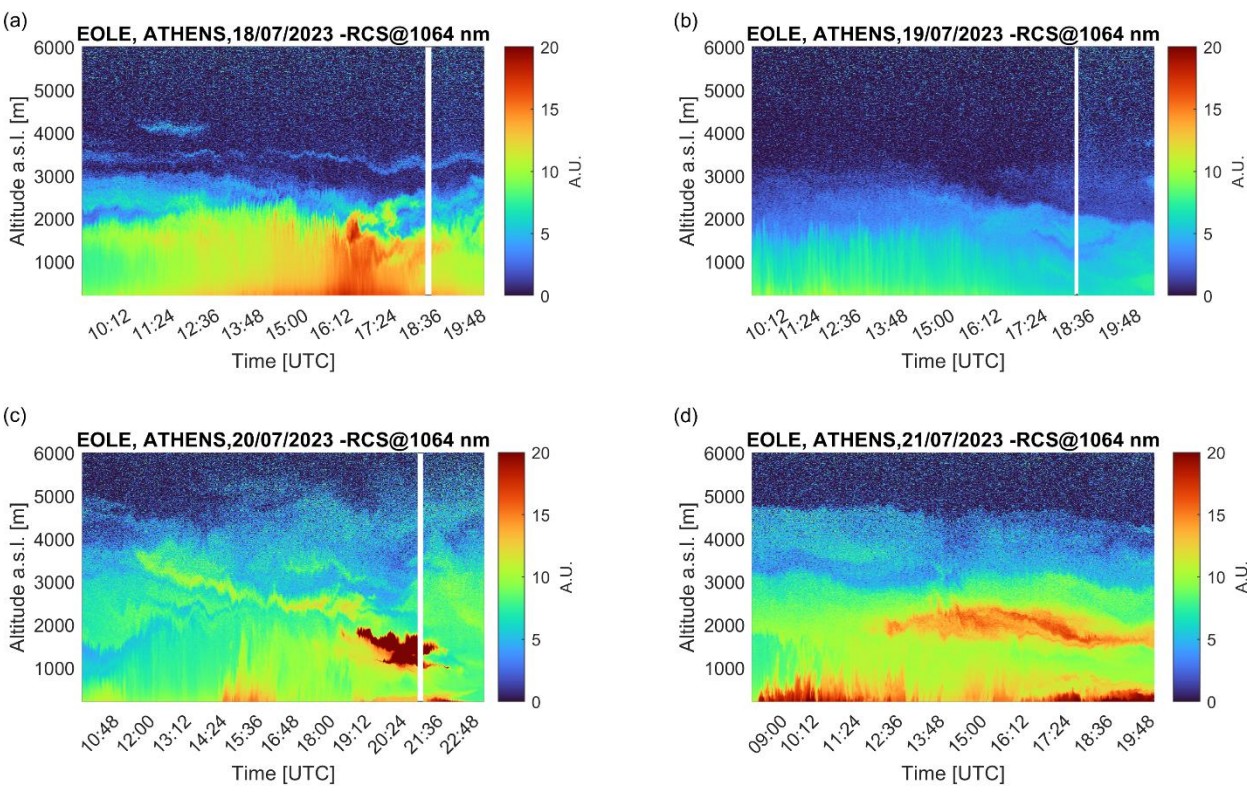

**Figure 4.** Spatio-temporal evolution of the range-corrected lidar signal (RCS) at 1064 nm for the period 18-21 July 2023.

On 21 July, the PBL extended up to ~1.5 km, above which a multi-layered structure developed: an elevated layer between 1.5
and 2.5 km, another from 2.5 to 3.0 km, and a thin lofted layer between 4.0 and 5.0 km, likely associated with long-range dust
transport. Between 12:40 and 18:00 UTC, a dense aerosol accumulation was noted around 2.5 km, while the higher layers
remained persistent. In the evening and nighttime hours, the atmospheric structure appeared to stabilize, with layers becoming
more stratified and less turbulent.



The vertical profiles of the aerosol optical properties retrieved on 21 July are presented in Figure 5, showing three distinct aerosol layers between 1.6 and 3.8 km. The lowest layer (1.62-2.16 km) is characterized by elevated aerosol backscatter coefficients, especially at 355 nm (3.79 Mm$^{-1}$ sr$^{-1}$) (Figure 5a), and a high AE (AEb$_{355/532}$ ≈ 1.60), indicating a dominance of fine-mode particles, likely biomass burning. However, the moderate PLDR (~0.11) suggests the presence of some non-spherical particles, possibly indicating dust and smoke mixtures (Sicard et al., 2012). The middle aerosol layer (2.52-3.00 km)

shows slightly lower backscatter but similarly high lidar ratios (LR$_{355}$ ≈ 36 sr), with an increase in PLDR to ~0.16, reinforcing the mixing hypothesis. The upper layer (3.18-3.84 km) presents the highest PLDR (~0.19) and the lowest AE, which is consistent with a more coarse-dominated aerosol layer, despite lower absolute backscatter and extinction values.

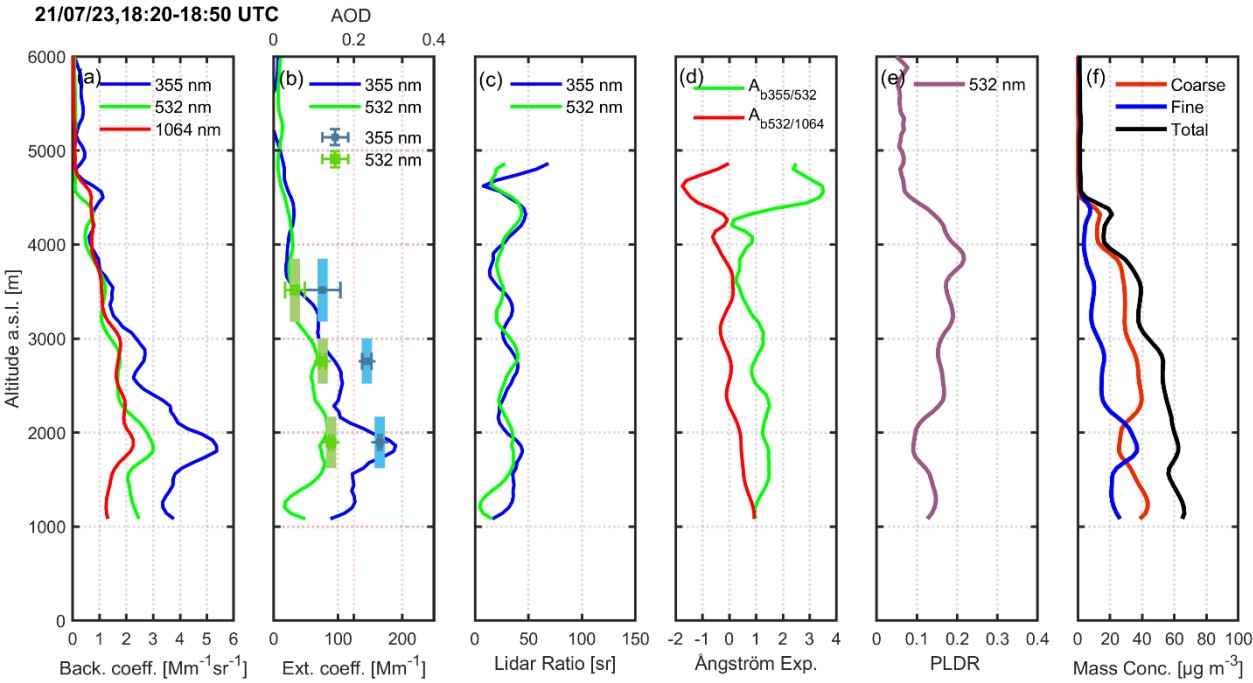

**Figure 5.** Vertical profiles of aerosol (a) backscatter coefficient (355, 532 and 1064 nm) (b) extinction coefficient and AOD (355 and 532
nm) (c) lidar ratio (355 and 532 nm) (d) Ångström exponent (e) PLDR (532 nm) and (f) aerosol mass concentration (coarse, fine mode and total particles) obtained by the EOLE and DEPOLE lidar systems on 21 July 2023 (18:20-18:50 UTC).

Figure 5b presents the aerosol extinction coefficients at 355 and 532 nm, which peak in the lower and middle layers, with values exceeding 100 Mm$^{-1}$ at 355 nm, further confirming a strong aerosol load. In addition, the layer-resolved AOD retrieved from the extinction lidar profiles highlights the significant optical contribution of each layer to the total column. Specifically,

the AOD at 355 nm ranged from 0.27 in the lower layers to 0.12 at higher altitudes, while at 532 nm it ranged from 0.14 to 0.05. These values are consistent with a mixed smoke-dust episode, with the higher AOD at shorter wavelengths indicating pronounced fine-mode particles from smoke, while the significant contribution at 532 nm reflects the presence of coarse mineral dust within the same air mass. Figure 5c shows lidar ratios, reaching 57 sr at 532 nm, which are typical of absorbing smoke but also possible dust contributions (Groß et al., 2011). Figure 5d presents the AE, showing a clear gradient from high



(smoke) in the lower layer to low (dust) in the upper layer (Baars et al., 2012; Salgueiro et al., 2021). Figure 5e illustrates the PLDR at 532 nm, which increases with height (PLDR ~0.2) and indicates the dust influence in the upper layers. Finally, Figure 5f presents the mass concentration profiles of fine, coarse, and total aerosols, highlighting the enhanced contribution from coarse-mode particles with increasing altitude. Within the identified aerosol layers, coarse-mode mass concentrations ranged from $20 \pm 7$ to $55 \pm 20 \, \mu g \, m^{-3}$, with values increasing up to 4.5 km. In contrast, the fine-mode mass concentration exhibited a

mean of $8 \pm 3 \, \mu g \, m^{-3}$, with a peak of $36 \pm 12 \, \mu g \, m^{-3}$ observed at around 1.8 km, corresponding to the lower aerosol layer. These vertical profiles confirm the presence of both fine and coarse particles, with the coarse mode clearly dominating at higher altitudes. The observed aerosol layers indicate a mixed event of biomass burning and Saharan dust over Athens. This is also supported by the FLEXPART simulations and lidar-derived optical properties (Figure S2(k-m)).

  Figure 6 presents time-series of columnar aerosol optical and microphysical properties retrieved by the NTUA CIMEL sun-

sky-lunar photometer for the period 18-21 July 2023, capturing the evolving impact of regional biomass burning and dust transport over Athens. Figure 6a shows the AOD at multiple wavelengths (340-1020 nm), along with the AE (440/870 nm). The elevated AOD values and relatively high AE (>1.5) observed during 18-19 July indicate the dominance of fine-mode particles, consistent with smoke from local and regional fires. On 20-21 July, a decrease in AE values along with sustained high AOD ones suggests a mixture of aerosols, due to the additional presence of coarse-mode dust particles from North Africa.





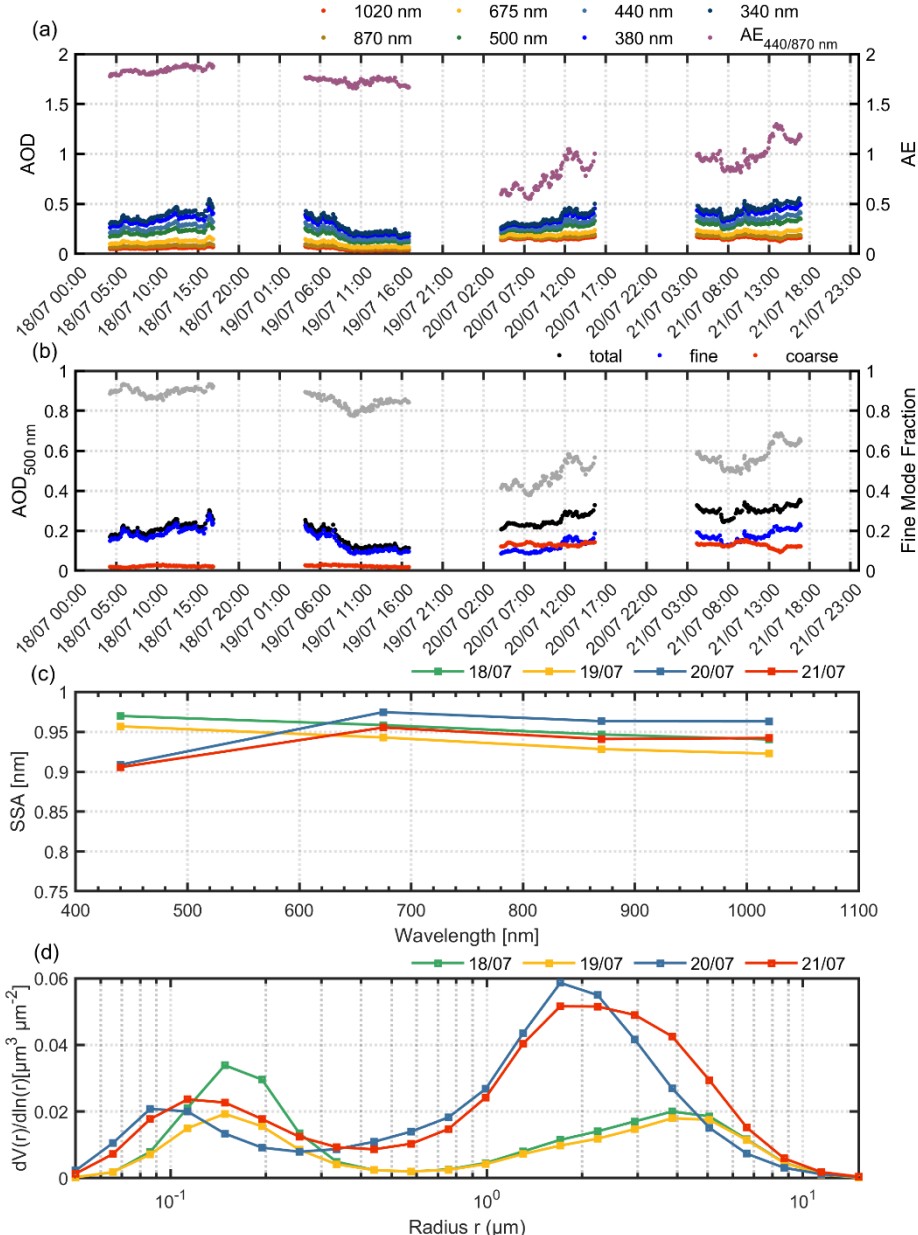

**Figure 6.** Time−series of values of the (a) AOD for various wavelengths (340 - 1020 nm) and AE at 440/870 nm (b) AOD at 500 nm along with the FMF (c) SSA wavelength dependence and (d) Particle volume size distribution obtained by the NTUA CIMEL sun-sky-lunar photometer for the period 18- 21 July 2023.

Figure 6b focuses on the AOD at 500 nm and the FMF values remained above 0.8 on 18-19 July, confirming the fine-mode

dominance typical of biomass-burning aerosols. A slight decline in FMF (< 0.6) on 20-21 July corresponds to the arrival of

Saharan dust, introducing a coarse-mode component to the aerosol mixture.





Figure 6c presents the wavelength dependence of SSA for the period 18-21 July 2023. The SSA values, reported at 440, 675, 870, and 1020 nm, show consistently high values (> 0.90), indicative of scattering-dominated aerosols. On 18 July, SSA values ranged from 0.97 at 440 nm to 0.94 at 1020 nm, suggesting a presence of fine-mode, weakly absorbing particles. A slight

decrease was observed on 19 July, indicating a modest increase in aerosol absorption Notably, on 20 July the SSA values were quite low at 440 nm (0.91), but higher values were found (~0.96) at longer wavelengths, possibly reflecting a mixture of absorbing fine-mode smoke and coarse-mode dust. By 21 July, the SSA spectrum remained relatively stable but slightly lower at shorter wavelengths, consistent with aged biomass burning aerosols that exhibit moderate absorption, particularly in the bluer region. Moreover, the volume size distribution (Figure 6d), showing a prominent fine-mode peak during 18-19 July,

which becomes broader on 20-21 July, reflecting the superposition of fine and coarse aerosol modes. This agrees well with the interpretation of a mixed aerosol event resulting from the interaction of smoke and Saharan dust over the Athens basin.

**3.4. Transport of fresh smoke particles (the case of 22-25 August 2023)**

Footprint emission sensitivities for BC from 22 to 25 August 2023 (12:00-18:00 UTC) (Figure 7) confirm that the air masses arriving over Athens between 0.5 and 4.0 km altitude passed over regions with intense wildfire activity, particularly over

northern Greece, and specifically the Rodopi and Evros regions (Figure 1c). This spatial overlap underscores a strong biomass burning (BB) influence on the aerosol composition observed over Athens during this episode.



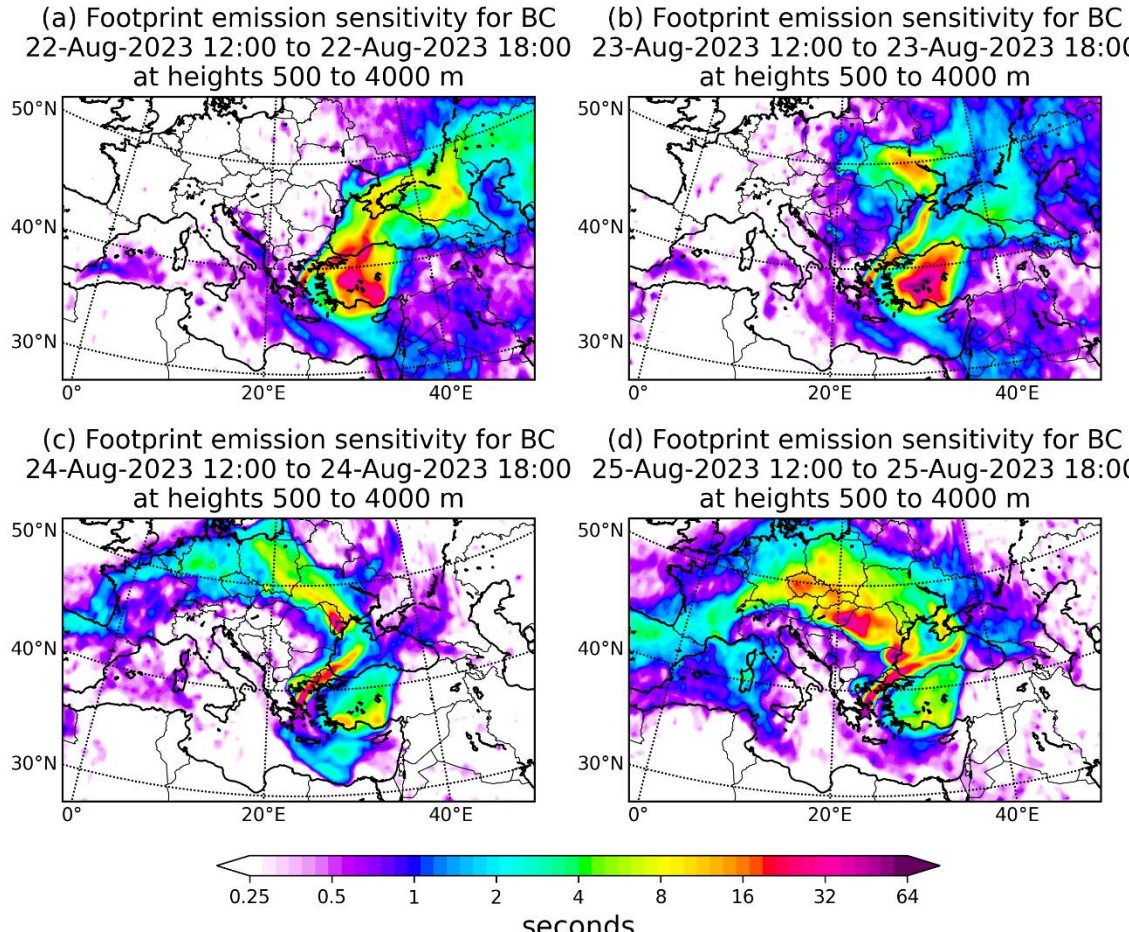

**Figure 7.** Footprint emission sensitivities for BC obtained using FLEXPART for the air masses arriving over Athens between 0.5 and 4.0 km from 22 to 25 August 2023 (12:00 UTC to 18:00 UTC).

More specifically, the footprint emission sensitivities from 22 to 25 August (Figure 7a-d) show increased residence time over the Evros and Rodopi wildfire zones (Figure 1b), suggesting direct transport of smoke plumes from forest fires. Additional sensitivity over parts of the Balkans and Turkey correspond to potential regional pollution contributions. In contrast, the lack of significant sensitivity over North Africa indicates a minimal dust influence during this period.

Figure 8 presents the spatio-temporal evolution of the range-corrected lidar signal (RCS) at 1064 nm over Athens from 22 to 455 25 August 2023, revealing persistent layers of transported biomass burning aerosols within the free troposphere and a dynamically evolving PBL.

On 22 August, the PBL extended up to ~1.0 km in the morning, with a distinct aerosol layer observed aloft between 3.0-4.0 km. Around 14:00-14:30 UTC, cloud formation occurred at ~4.0 km, likely associated with the interaction of lofted BB aerosols and residual moisture. Between 14:30 and 16:00 UTC, this upper layer gradually descended to ~2.0 km, and persisted



at that height until 19:00 UTC, positioned above a slightly deepened PBL (~1.3 km). A diffuse aerosol layer persisted above, extending up to 4.0 km height.

On 23 August, a shallow PBL (≤1.0 km) was maintained throughout the day. Overlying this, persistent lofted aerosol layers were identified between 2.0-3.0 km and 3.0-4.0 km, intensifying progressively through the evening. Cloud formation was observed at ~5.0 km, attributed to BB plume-driven condensation activity.

During 24 August, multiple stratified aerosol layers were present above the PBL (~1.0 km), notably at 2.0-3.0 km, 3.2-3.5 km, and 4.0-4.8 km as early as 11:45 UTC. These layers became progressively more optically dense during the day. Between 15:30 and 19:20 UTC, mid-tropospheric cloud formation occurred at ~1.8 km and ~2.8 km, likely due to interactions between hygroscopic smoke particles and elevated humidity (Figure S5). The elevated aerosol layering persisted through the night (after 20:00 UTC), with continued cloud presence.

On 25 August, the lidar signal revealed a relatively stable vertical aerosol structure, with a well-defined layer up to ~2.0 km and a distinct elevated layer near ~4.0 km. The upper layer intensified in the late afternoon (around 17:00 UTC), resulting in cloud formation by 20:00 UTC, consistent with sustained BB aerosol presence and favorable thermodynamic conditions.

These observations confirm the persistent influence of BB aerosols over Athens during this period, affecting both aerosol vertical structure and cloud formation processes.



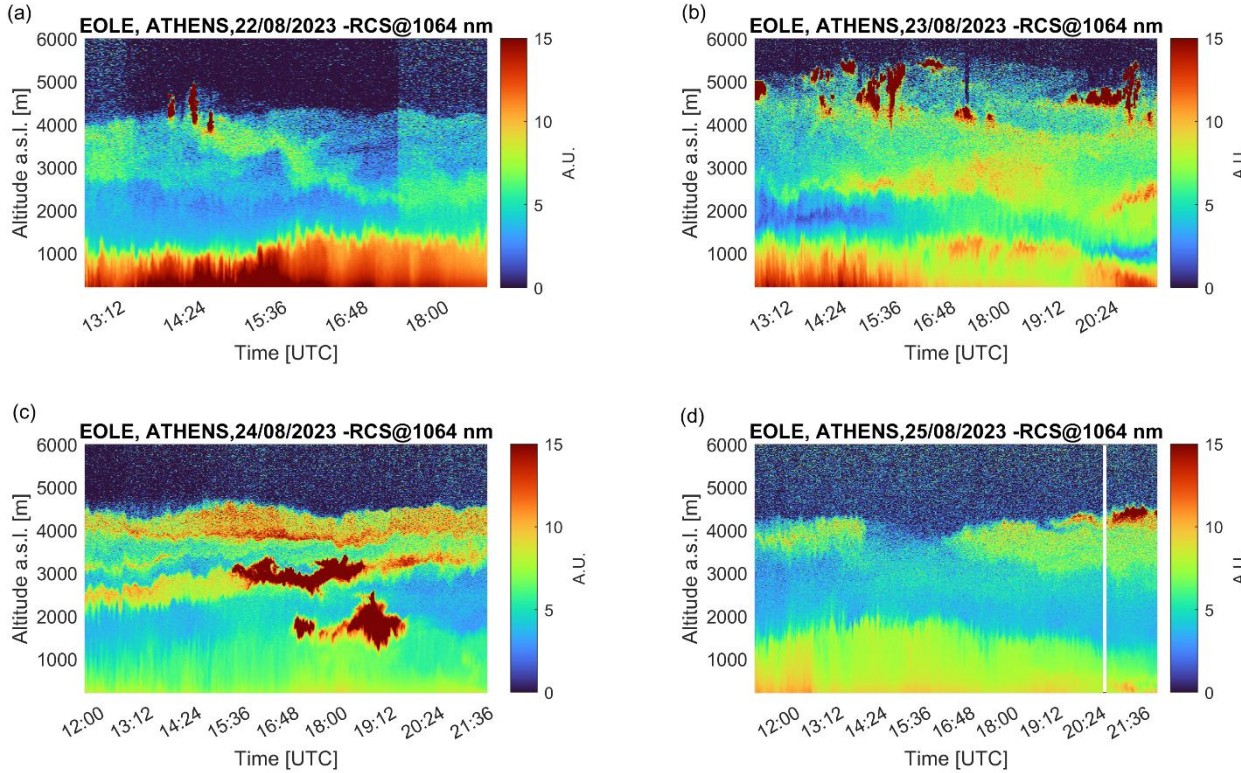

**Figure 8.** Spatio-temporal evolution of the range-corrected lidar signal (RCS) at 1064 nm for the period 18-21 August 2023.

Figure 9 illustrates the vertical profiles of the aerosol optical properties retrieved on 24 August 2023, during an episode dominated by biomass burning smoke particles transported from the Evros and Rodopi regions in northeastern Greece, as confirmed by FLEXPART simulations (Figure 7). The backscatter coefficients (Figure 9a) reveal two distinct aerosol layers between 3.0-3.72 km and 3.96-4.62 km. The upper layer exhibited enhanced backscatter coefficients at all 3 wavelengths (e.g. $b_{355} \approx 3.25$ Mm$^{-1}$ sr$^{-1}$), suggesting a more concentrated aerosol load, while both layers showed backscatter AE ($Ab_{355/532} \approx 1.4$) indicative of fine-mode particles.

Figure 9b presents the aerosol extinction coefficients at 355 and 532 nm, peaking up to 130 Mm$^{-1}$ in the lower layer and 120 Mm$^{-1}$ in the upper one. These high extinction values point to substantial aerosol optical depth, consistent with a dense smoke plume. Furthermore, the layer-resolved AOD at 355 nm, ranged from 0.32 in the lower, more concentrated smoke layer to 0.06 in the upper one, while at 532 nm it ranged from 0.12 to 0.0035. The much larger AOD at the shorter wavelength reflects the dominance of fine-mode smoke particles, which exhibit a strong wavelength dependence with rapidly decreasing optical depth across longer wavelengths. The extremely low AOD values at 532 nm aloft also indicate that the upper layers contained more diluted smoke, with only a minor contribution to the total column Moreover, the lidar ratios of 51 sr (355 nm) and 85 sr (532



nm) in the lower layer, and slightly reduced values (50 sr and 65 sr) in the upper layer (Figure 9c) are both indicative of the

presence of biomass burning aerosol (Alados-Arboledas et al., 2011; Müller et al., 2007).

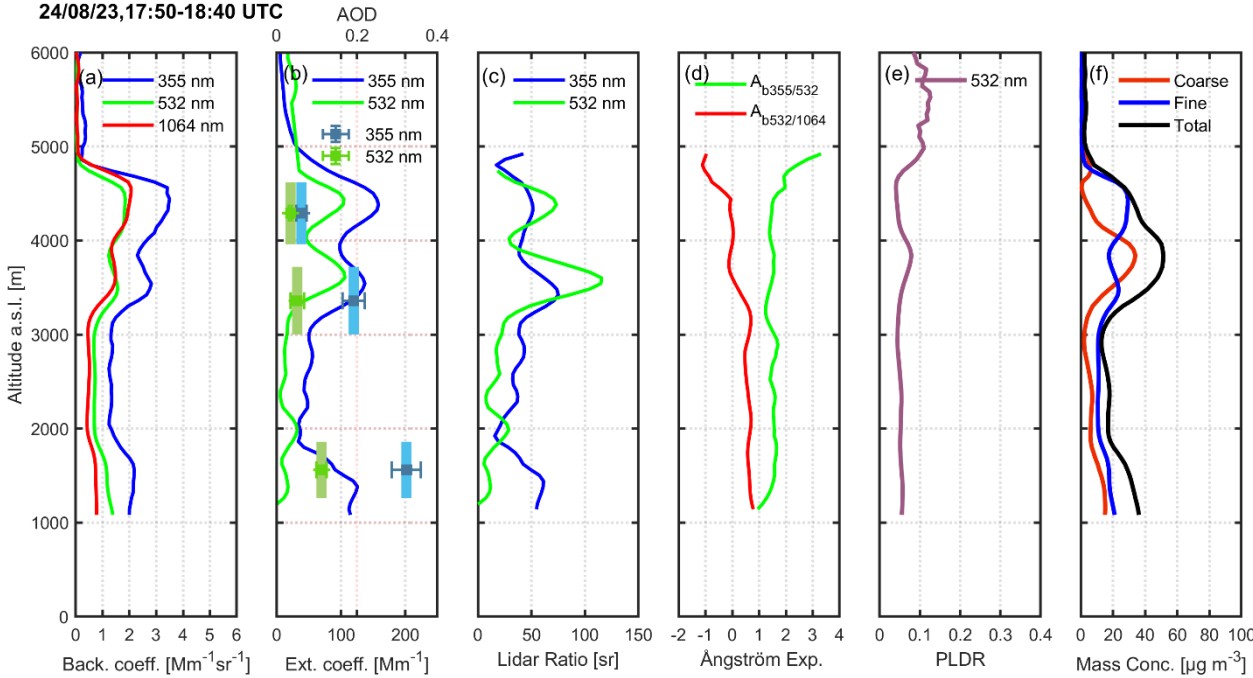

**Figure 9.** Vertical profiles of aerosol (a) backscatter coefficient (355, 532 and 1064 nm) (b) extinction coefficient and AOD (355 and 532 nm) (c) lidar ratio (355 and 532 nm) (d) Ångström exponent (e) PLDR (532 nm) and (f) aerosol mass concentration (coarse, fine mode and
total particles) obtained by the EOLE and DEPOLE lidar systems on 24 August 2023 (17:50-18:40 UTC).

The aerosol Ångström exponents, $Ab_{355/532} \approx 1.4$ and $Ab_{532/1064}$ ranged from 0.2 to −0.05, the latter suggesting the presence of

slightly larger particles (Nicolae et al., 2013) in the upper layer. The low values of the PLDR ($\leq 0.057$) (Figure 10e), further

supported the absence of mineral dust and the predominance of spherical smoke particles (Gidarakou et al., 2024; Haarig et

al., 2019). Finally, Figure 9f presents the vertical profiles of aerosol mass concentration, clearly showing fine-mode dominance

across both elevated layers, consistent with the presence of biomass burning aerosols, as also supported by FLEXPART

simulations. Fine-mode mass concentrations ranged from $23 \pm 7$ to $30 \pm 10 \, \mu g \, m^{-3}$ within the layers, with increasing values

observed up to 4.5 km. These elevated concentrations are attributed to biomass-burning emissions from intense wildfires in

Evros and Rodopi, with the smoke plume clearly visible in the MODIS/Aqua satellite image acquired on the afternoon of 22

August 2023 (Figure 1c). A slight contribution from coarse-mode particles was also detected, but the overall signal remained

dominated by the fine fraction, typical of aged, lofted smoke layers.

Figure 10 presents the time series of the columnar aerosol optical and microphysical properties retrieved from the NTUA

CIMEL sun-sky-lunar photometer during the 22-25 August 2023 period, which coincided with intense wildfire activity in

northern Greece. The AOD (Figure 10a) remained elevated throughout the period, particularly at shorter wavelengths,

specifically it increased from ~0.3 to 0.5 the first two days, while it increased a lot on 24 August reaching values up to 1.5



indicating the arrival of intense smoke plumes. The AE values were consistently high from 1.5-2.0 with peak again on 24 August confirming the dominance of fine-mode aerosols, biomass-burning smoke. Moreover, the AOD at 500 nm had the same trend and remained peaked on 24 August reaching ~0.7, with the FMF (Figure 10b) exceeding 0.6 on 22 August reaching even 1.0 on the last day, further supporting the strong influence of fine particles. Figure 10c shows the SSA as a function of wavelength from AERONET observations between 22 and 25 August 2023, highlighting variations in aerosol absorption and

scattering properties across four wavelengths: 440, 675, 870, and 1020 nm. On 22 August, the SSA values were moderate ($\approx 0.92$-0.94), suggesting internally mixed aerosols with moderate absorption, likely linked to transported smoke. A peak in SSA values was observed on 23 August, reaching 0.98 at 440 nm, indicating highly scattering aerosols, consistent with a cleaner or more aged smoke plume. In contrast, 24 August showed a marked decrease in SSA values with wavelength from 0.93 at 440 nm to just 0.85 at 1020 nm was found on 24 August, revealing the presence of more absorbing, coarse-mode

particles, likely due to a denser biomass burning plume (Figure 10c). By 25 August, the SSA values increased again (up to 0.96 at 440 nm), suggesting a shift back toward less absorbing, finer-mode aerosols. This spectral dependence suggests the presence of internally mixed carbonaceous particles, with stronger absorption in the visible and near-infrared range (Russell et al., 2010).

Finally, the particle volume size distribution shown in Figure 10d is characterized by a dominant fine-mode peak, accompanied

by a noticeable coarse-mode contribution, consistent with the presence of smoke plumes from regional wildfires influencing the air masses over Athens during this period.





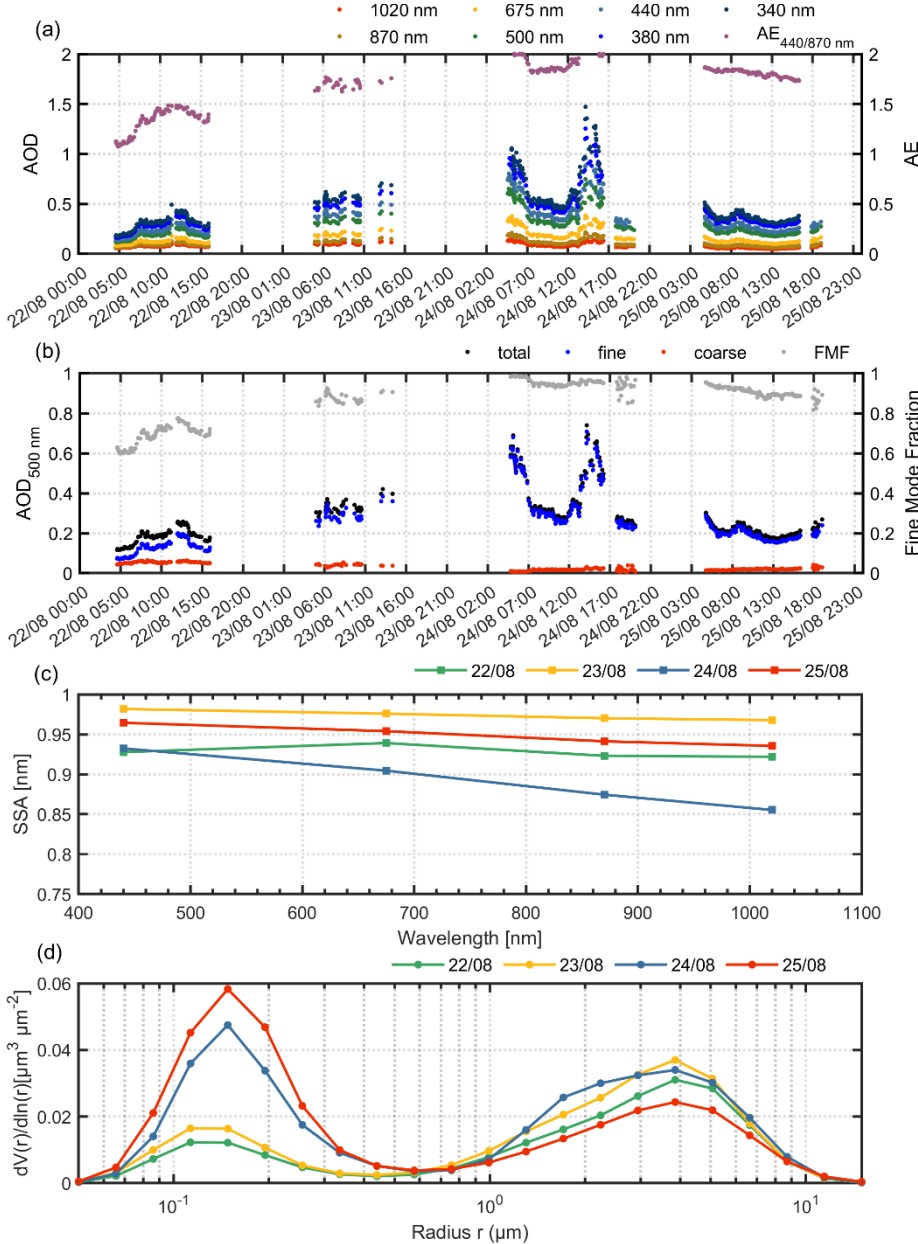

**Figure 10.** Time−series of values of the (a) AOD for various wavelengths (340 - 1020 nm) and AE at 440/870 nm (b) AOD at 500 nm along with the FMF (c) SSA wavelength dependence and (d) particle volume size distribution obtained by the NTUA CIMEL sun-sky-lunar photometer for the period 22- 25 August 2023.

## 3.6 Aerosol optical and microphysical properties

Figure 11 presents the geometrical (Thickness and Base and Top of the aerosol layers) and optical (backscatter coefficient,

LR, PLDR and microphysical properties (effective radius and refractive index - real and imaginary) for the period from 17





July to 30 August, according to aerosols layers observed. During the biomass burning and dust period (17 July to 21 July), aerosol layers were observed from heights of 0.96 to 5.34 km, while the mean thickness of the aerosol layers was equal to 0.60 km. High values of aerosol backscatter coefficient that were recorded during this period, exceeding $12 \pm 4$ Mm$^{-1}$ sr$^{-1}$ at 355 nm (20 July), as well as enhanced ones at 532 nm, even reaching $6.8 \pm 2$ Mm$^{-1}$ sr$^{-1}$, are indicative of dense aerosol loads. The

LR at 355 nm varied from $30 \pm 6$ to $86 \pm 14$ sr (21 July), presenting an average value equal to $48 \pm 4$ sr. The LR at 532 nm presented an increased range from 27 (18 July) to 58 sr (21 July) and an average value of $37 \pm 8$ sr. These LR values indicate the significant presence of Saharan dust aerosols, along with fresh biomass burning particles (Mylonaki et al., 2021; Papagiannopoulos et al., 2018). Furthermore, PLDR values at both 355 and 532 nm ranged approximately from 0.03 to 0.19, with a mean value of $0.06 \pm 0.01$ at 355 nm and $0.08 \pm 0.01$ at 532 nm. These PLDR values at 355 and 532 nm highlight the

presence of non-spherical particles (0.03 - 0.08), with slightly enhanced values indicating smoke-dust mixtures (PLDR ~0.1-0.2). Specifically, the PLDR was stronger in lower layers and reduced in upper layers, supporting mixed-type aerosols (Giannakaki et al., 2016; Nemuc et al., 2013). variability and differing sensitivity to aerosol layers. The peak of $R_{eff}$ values on 21 July reflects the presence of a dense dust plume mixed with smoke particles (Weinzierl et al., 2011). Real refractive index - mR ranged from $1.53 \pm 0.05$ to $1.61 \pm 0.05$, while imaginary parts - mI reached up to $0.0093 \pm 0.005$, especially on 18 July,

indicating the presence of moderately absorbing, smoke particles (Nicolae et al., 2013).

During the period from 22 to 25 August, aerosol layers were observed between ~1.0 and 5.5 km, with an average thickness of 0.56 km. Mean aerosol backscatter coefficients reached $3.03 \pm 0.44$ Mm$^{-1}$ sr$^{-1}$ at 355 nm and $1.65 \pm 0.24$ Mm$^{-1}$ sr$^{-1}$ at 532 nm, with peak values of 5.64 and 3.42 Mm$^{-1}$ sr$^{-1}$, respectively, indicating the presence of a moderately dense aerosol load. The LR values at 355 nm ranged from ~25 to 75 sr, with an average of $44 \pm 11$ sr, while LR at 532 nm varied from ~15 to 52

sr, averaging $32 \pm 13$ sr. These values suggest a mixture of aerosol types, likely dominated by fresh biomass burning particles.




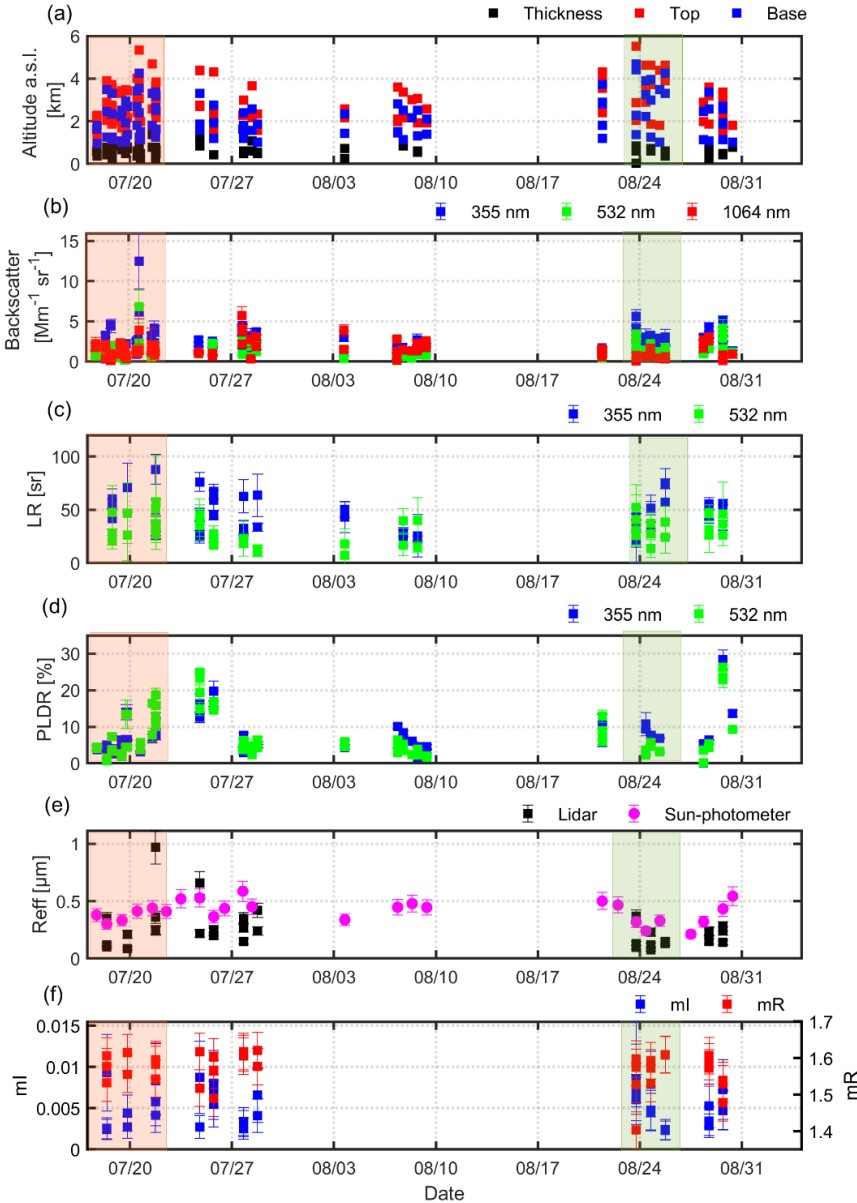

**Figure 11**. Aerosol geometrical, optical, and microphysical properties retrieved over Athens during the period from 17 July to 30 August 2023: (a) base, top, and thickness off the aerosol layers, (b) backscatter coefficients (355, 532, and 1064 nm), (c) LR (355 and 532 nm), (d) PLDR (355 and 532 nm), (e) Reff, and (f) imaginary (mI) and real (mR) parts of the complex refractive index.


The $LR_{532}/LR_{355}$ ratio had a mean of ~0.72, further supporting this classification (Nicolae et al., 2013). PLDR values at 355 and 532 nm ranged from ~0.04 to 0.10, with average values of 0.08 and 0.04, respectively, indicating the



presence of mostly mixed non-spherical particles with smoke ones. In terms of microphysical properties, the retrieved $R_{eff}$ values ranged from ~0.07 to 0.37 μm, with an average of 0.16 ± 0.02

μm. The sun photometer-derived Reff values ranged from 0.24 to 0.47 μm, averaging 0.34 μm. The real part of the refractive index (mR) varied from 1.43 to 1.61 (mean: 1.56), and the imaginary part (mI) reached up to 0.0103, indicating strongly absorbing aerosols (Weinzierl et al., 2011).

During the biomass burning and dust period (17 July to 21 July), aerosol layers were observed from heights of 0.96 to 5.34 km, while the mean thickness of the aerosol layers was equal to 0.60 km. High values of aerosol backscatter coefficient that

were recorded during this period, exceeding $12 \pm 4$ Mm$^{-1}$ sr$^{-1}$ at 355 nm (20 July), as well as enhanced ones at 532 nm, even reaching $6.8 \pm 2$ Mm$^{-1}$ sr$^{-1}$, are indicative of dense aerosol loads. The LR at 355 nm varied from $30 \pm 6$ to $86 \pm 14$ sr (21 July), presenting an average value equal to $48 \pm 4$ sr. The LR at 532 nm presented an increased range from 27 (18 July) to 58 sr (21 July) and an average value of $37 \pm 8$ sr. These LR values indicate the significant presence of Saharan dust aerosols, along with fresh biomass burning particles (Janicka et al., 2017; Murayama et al., 2004). Furthermore, PLDR values at both 355 and 532

nm ranged approximately from 0.03 to 0.19, with a mean value of 0.06 ± 0.01 at 355 nm and 0.08 ± 0.01 at 532 nm. These PLDR values at 355 and 532 nm highlight the presence of non-spherical particles (0.03 - 0.08), with slightly enhanced values indicating smoke-dust mixtures (PLDR ~0.1-0.2). Specifically, the PLDR was stronger in lower layers and reduced in upper layers, supporting mixed-type aerosols (Giannakaki et al., 2016; Nemuc et al., 2013).

Regarding the retrieved aerosol microphysical properties, the $R_{eff}$ ranged from ~0.087 ± 0.01 to 0.97 ± 0.14 μm on 21 July,

with an average of 0.4 ± 0.05 μm. Sun-sky-lunar photometer values reached up to 0.44 ± 0.07 μm, averaging 0.37 ± 0.05 μm, indicating internal variability and differing sensitivity to aerosol layers. The peak of $R_{eff}$ values on 21 July reflects the presence of a dense dust plume mixed with smoke particles (Weinzierl et al., 2011). Real refractive index - mR ranged from 1.53 ± 0.05 to 1.61 ± 0.05, while imaginary parts - mI reached up to 0.0093 ± 0.005, especially on 18 July, indicating the presence of moderately absorbing, smoke particles (Nicolae et al., 2013).

During the period from 22 to 25 August, aerosol layers were observed between ~1.0 and 5.5 km, with an average thickness of 0.56 km. Mean aerosol backscatter coefficients reached 3.03 ± 0.44 Mm$^{-1}$ sr$^{-1}$ at 355 nm and 1.65 ± 0.24 Mm$^{-1}$ sr$^{-1}$ at 532 nm, with peak values of 5.64 and 3.42 Mm$^{-1}$ sr$^{-1}$, respectively, indicating the presence of a moderately dense aerosol load. The LR values at 355 nm ranged from ~25 to 75 sr, with an average of 44 ± 11 sr, while LR at 532 nm varied from ~15 to 52 sr, averaging 32 ± 13 sr. These values suggest a mixture of aerosol types, likely dominated by fresh biomass burning particles.

The $LR_{532}/LR_{355}$ ratio had a mean of ~0.72, further supporting this classification (Nicolae et al., 2013). PLDR values at 355 and 532 nm ranged from ~0.04 to 0.10, with average values of 0.08 and 0.04, respectively, indicating the presence of mostly mixed non-spherical particles with smoke ones. In terms of microphysical properties, the retrieved $R_{eff}$ values ranged from ~0.07 to 0.37 μm, with an average of 0.16 ± 0.02 μm. The sun photometer-derived Reff values ranged from 0.24 to 0.47 μm, averaging 0.34 μm. The real part of the refractive

index (mR) varied from 1.43 to 1.61 (mean: 1.56), and the imaginary part (mI) reached up to 0.0103, indicating strongly absorbing aerosols (Weinzierl et al., 2011).



## 3.7 Impact of aerosols on ground solar UVB irradiance

Figure 12 (a-d) show the measured UVB irradiance at 320 nm (blue line) from the Brewer spectrophotometer at specific SZAs

of 60°, 50°, 40°, and 30°, respectively, compared with UVB estimates from the clear-sky radiative transfer model (LibRadtran;

red line). The percentage difference between measurements and model values is also shown (black dashed line). Figure 12 (e-

g) present the diurnal evolution of UVB irradiance at 320 nm for three characteristic cases: (e) smoke + dust (21 July), (f) dust

only (23 July), and (g) smoke only (24 August). Measured UVB (blue) is plotted alongside clear-sky model simulations (red)

to highlight the attenuation impacts of different aerosol types on surface UVB levels.

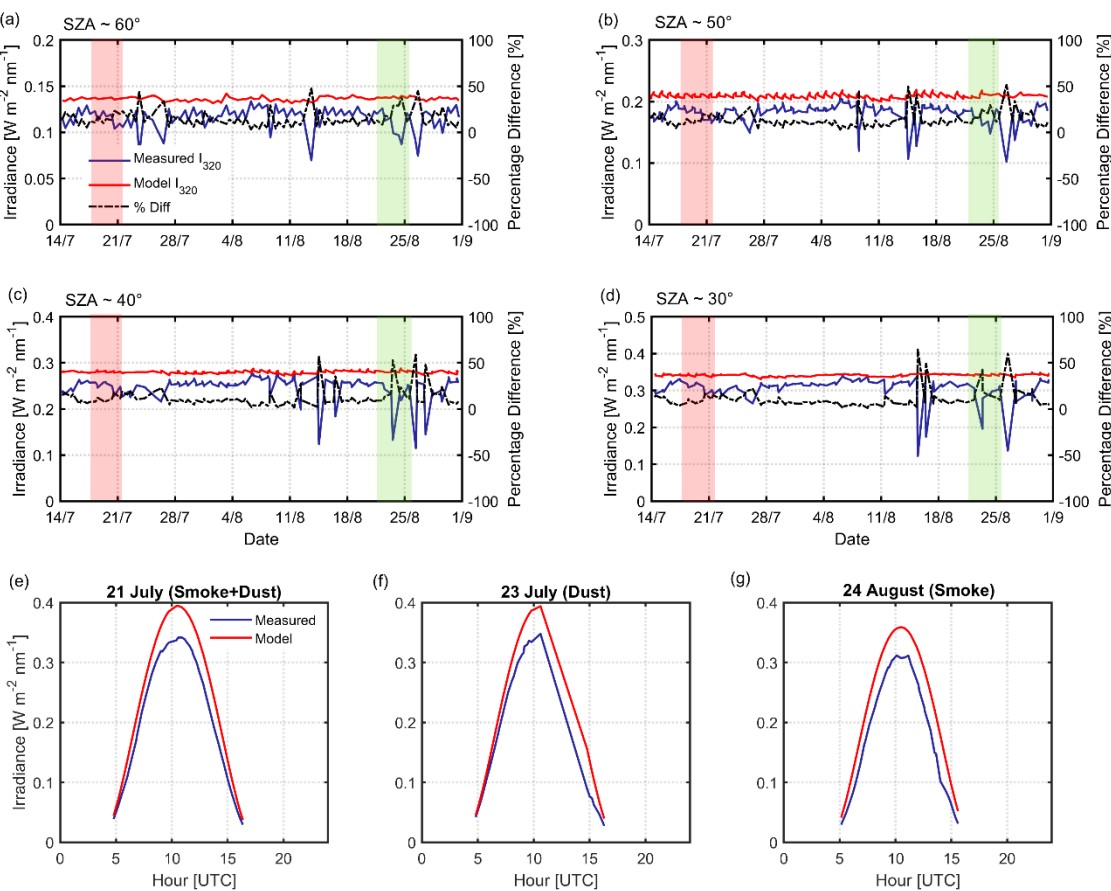

**Figure 12.** UVB irradiance attenuation at 320 nm for the period 14 July-30 August at SZAs of (a) 60°, (b) 50°, (c) 40°, and (d) 30°, as
measured by the Brewer spectroradiometer (blue) and compared with clear-sky simulations from the libRadtran model (red). The percentage
difference between observations and model is also shown (black dashed line), as well as the diurnal UVB irradiance at 320 nm for three
representative cases: (e) 21 July (smoke + dust), (f) 23 July (dust), and (g) 24 August (smoke), compared to clear-sky values.

Overall, the largest discrepancies are observed during the second case study (22-25 August), when biomass burning aerosols

were dominant. In particular, on 24 August, the percentage difference at SZA = 60° reached up to 40% (Figure 12a), while at

SZA = 30°, a peak difference of 42% was recorded on 23 August (Figure 12d), highlighting the strong attenuation effect of





smoke. In contrast, during the first case study (18-21 July), which was influenced by a mixture of dust and smoke, UVB attenuation was less pronounced. On 21 July, for example, the difference at SZA = 60° reached ~23%, and up to 11% at SZA = 40° (Figure 12c). A similar pattern is observed for dust aerosol conditions on 23 July. However, when smoke is the dominant

aerosol type, as in late August, the UVB attenuation becomes significantly stronger, especially at lower SZAs.

These patterns are also supported by the diurnal plots in Figure 12 (e-g), which present UVB irradiance measurements at 320 nm for three selected days, each representing a different aerosol event discussed in Section 3: 21 July (smoke + dust), 23 July (dust-dominated), and 24 August (smoke-dominated). On 21 July, elevated AOD resulted from the combined presence of smoke and dust, while 23 July was characterized by very high AOD due to dust. The most extreme UVB attenuation occurred

on 24 August, coinciding with the highest AOD values of the period, driven by smoke. In contrast, the least attenuation was observed on 23 July. These differences align with aerosol optical characteristics shown in Figure 2: 21 July featured high Ångström exponent (AE > 1) and fine-mode fraction (FMF ≈ 0.6), indicating mixed aerosols, 23 July had low AE (~0.5) and FMF (~0.3), consistent with coarse-mode dust dominance, while 24 August exhibited strong fine-mode influence, with AE around 2.0 and AOD at 340 nm exceeding 1.0.

The SSA in the UV can differ significantly from the SSA at visible wavelengths. By comparing the measured spectra with modelled spectra, we estimated the SSA at 320 nm, as well as the SSA for irradiances that are within the uncertainty range of the Brewer spectral measurements (i.e., 5%) (Bais et al., 2001). At Figure 13, we show the SSA values for the same days as in Fig. 12 (e) - (g). For the smoke event (24 August) the SSA at 320 nm is similar to the SSA at 440 nm (Fig. 10). Very low SSA values (0.75 - 0.8), well below the SSA values in the visible spectral region (Fig. 6) were found for the dust dominated mixture

(23 July), which is in agreement with the results of past studies (Fountoulakis et al., 2019; Raptis et al., 2018). Even assuming irradiance values that are 5% above the measured irradiances, results in this case in SSA values are below 0.9.

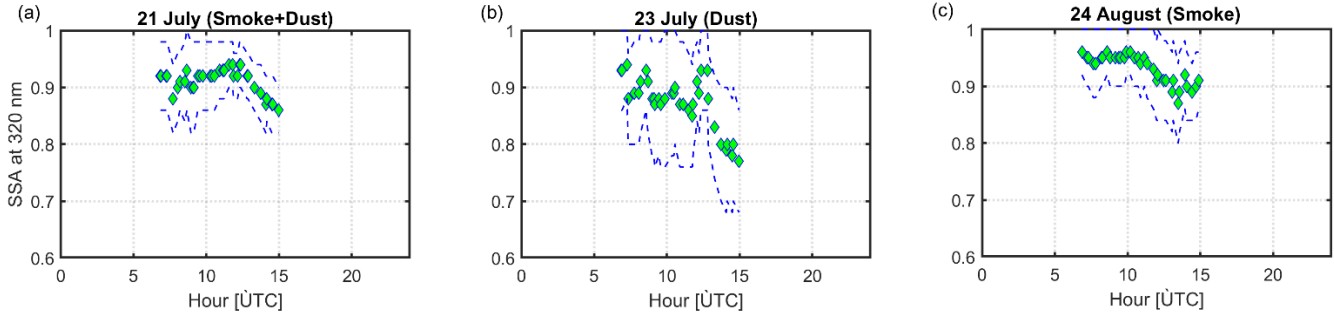

**Figure 13.** SA at 320 nm for three representative cases: (a) 21 July (smoke + dust), (b) 23 July (dust), and (c) 24 August (smoke). The dotted
blue lines represent the SSA values for irradiances that are within ± 5% around the measured values.





## 4. Summary and conclusions

Summer 2023 was marked one of the most severe wildfire seasons in recent Greek history, with large-scale fires affecting regions, such as Evros, Rodopi, Attica, the Peloponnese, and several islands. This study examined the optical and microphysical properties of smoke aerosols (sometimes mixed with dust particles) long-ranged transported over Athens from two major wildfires in Greece (18-21 July and 22-25 August 2023) and investigated the aerosol impact on UVB solar radiation at ground level using a combination of satellite observations (MODIS), FLEXPART transport modelling, ground-based lidar and sun-sky-lunar photometer measurements, and in situ aerosol and radiation data.

Throughout the observation period (17 July-31 August), Athens experienced significant aerosol loading, largely attributed to biomass burning events. The most intense air pollution episode occurred in late August, and was associated with wildfires in northeastern Greece. Elevated AOD values, consistently high Ångström exponent (AE > 1), and fine-mode fraction (FMF > 0.9 on 24 August) indicated the dominant presence of fine, absorbing particles, typical of fresh smoke. AOD values retrieved from Brewer and CIMEL instruments (in the 320 - 1020 nm spectral region) showed close agreement (~0.05 difference), supporting the reliability of the optical data. Ground level chemical composition analyses revealed peak organic aerosol concentrations (22.77 µg m$^{-3}$ on 22 August), with black carbon fraction from biomass burning exceeding 80%. $PM_{10}$ and $PM_{2.5}$ concentrations were also enhanced, reaching hourly maxima of 94 and 49 µg m$^{-3}$, respectively, with $PM_{2.5}/PM_{10}$ ratios up to 0.7, further confirming the presence of fine aerosols from biomass burning processes.

During the July episode (17 - 21 July), our lidar observations showed elevated aerosol layers between 0.96 and 5.34 km, with an average thickness of 0.60 km. High values of aerosol backscatter coefficients were recorded, along with LR values of $48 \pm 4$ sr (355 nm) and $37 \pm 8$ sr (532 nm), consistent with smoke-dust mixtures. PLDR values of 0.06 (355 nm) and 0.08 (532 nm) further supported our findings. Aerosol microphysical retrievals indicated effective radii up to 0.97 µm, real refractive indices between 1.53-1.61, and imaginary parts up to 0.0093, suggesting moderately absorbing particles. In contrast, during 22-25 August, the aerosol layers extended up to 5.5 km with slightly thinner mean thickness (0.56 km). Aerosol backscatter coefficient values were lower in comparison to the first case (18 - 21 July), but the LRs remained elevated (around 45 sr). PLDR values averaged at 0.08 (355 nm) and 0.04 (532 nm), while the retrieved particles values showed smaller effective radii (mean: 0.16 µm), with higher imaginary refractive index (0.0103) values, indicating the presence of strongly absorbing particles (Pokhrel et al., 2016).

The impact of the aerosol particles on solar UV radiation at ground level was substantial. Indeed, the UVB irradiance dropped by up to 50% (26 August) during the smoke-heavy days compared to the previous ones. The August episode showed even stronger attenuation, with UVB reductions of 40% at SZA = 60° and 42% at SZA = 30°. In contrast, the mixed dust-smoke period in July showed smaller reductions (~25%). Daily UVB cycles supported these trends: 21 July exhibited moderate attenuation alongside AE > 1 and FMF ≈ 0.6; 23 July (dust-dominated) showed weaker UVB effects with AE ~0.5 and FMF ~0.3; while 24 August (smoke-dominated) recorded AE ~2.0, AOD > 1.0, and the strongest UVB suppression. During the smoke events the absorption efficiency of the particles was found to be comparable or lower than the absorption efficiency in



the visible wavelengths (i.e., comparable or larger SSA in the UV), while the presence of dust results in lower SSA values in
the UVB relative to the visible region.

The FLEXPART simulations confirmed the transport of biomass burning aerosols from northeastern Greece regions (Evros, Rodopi) to Athens. Additional contributions from Saharan dust and regional pollution, particularly in July, modulated the observed aerosol properties.

Overall, the present study highlights the value of integrating satellite, modelling, and ground-based observations to assess the
complex impacts of wildfire aerosols. Our findings highlighted the strong influence of fire emissions on aerosol optical, microphysical, and radiative properties over Athens in summer 2023. In addition, we provided a systematic comparison between Brewer-derived AOD and CIMEL AERONET AOD, showing good agreement (average difference < 0.05, standard deviation 0.05-0.1 depending on wavelength). This further validates the use of Brewer measurements for investigating aerosol effects on UVB radiation at ground level, since UVB radiation near ground plays a crucial role in human health and local air
pollution photochemistry.

*Author contributions.* Conceptualization, A.P. and M.G.; methodology, M.G. and A.P.; software, M.G., I.V., N.E., C.G.Z, E.K. and M.M; validation, I.V., M.G., M.I.G., O.Z., K.G., K. Eleftheriadis., E.D., K. Eleftheratos, and I.F.; investigation, M.G., A.P., K. Eleftheratos. and I.F.; data curation, M.G., E.D., O.Z., M.M., I.V., I.F.; writing—original draft preparation, M.G.;
writing—review and editing, M.G., A.P., K. Eleftheratos and I.F.; visualization, M.G., N.E., C.G.Z.; All authors have read and agreed to the published version of the manuscript.

*Acknowledgements.* The authors acknowledge the use of data and/or imagery from NASA's Fire Information for Resource Management System (FIRMS) (https://earthdata.nasa.gov/firms), part of NASA's Earth Observing System Data and
Information System (EOSDIS). The AERONET project at NASA GSFC is supported by the Earth Observing System Program Science Office Cal-Val, Radiation Science program at NASA headquarters, and various field campaigns. We would also like to thank AERONET for their continuous efforts in providing high-quality measurements and derivative products.

*Financial support.* This article is funded from the COST Action Harmonia, CA21119, supported by COST (European
Cooperation in Science and Technology). M.G. was supported by the Hellenic Foundation for Research and Innovation (HFRI) under the 4th Call for HFRI Ph.D. Fellowships (Fellowship number: 9293). The project 21GRD02 BIOSPHERE has received funding from the European Partnership on Metrology, cofinanced by the European Union's Horizon Europe Research and Innovation Programme and by the Participating States. The FLEXPART results used a virtual access service that is supported by the European Commission under the Horizon 2020—Research and Innovation Framework Programme, H2020-INFRAIA-
2020-1, ATMO-ACCESS. Grant Agreement number: 101008004. The computations/simulations/[SIMILAR] were performed using resources provided by Sigma2—the National Infrastructure for High-Performance Computing and Data Storage in Norway.



*Competing interests.* The authors declare that they have no conflict of interest.


*Data availability.* Lidar, and air pollution data are available upon request from the corresponding author (alexandros.papagiannis@epfl.ch). The data are not publicly available as they are part of a larger dataset which is not published yet. All AERONET data used in this work can be accessed through the AERONET web page: http://aeronet.gsfc.nasa.gov/. All FLEXPART results can be viewed or downloaded from https://atmo-

access.nilu.no/NTUA_2023.py. The air mass backward trajectory analysis is based on air mass transport computation by the NOAA (National Oceanic and Atmospheric Administration) HYSPLIT (HYbrid Single-Particle Lagrangian Integrated Trajectory) model (http://ready.arl.noaa.gov/HYSPLIT_traj.php). GDAS1 (Global Data Assimilation System 1) re-analysis products from the National Weather Service's National Centers for Environmental Prediction are available at https://www.ready.noaa.gov/ gdas1.php.

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
