# Peer review of "Exceptional wildfire smoke over Greece in summer 2023: a synergistic study of aerosol optical-microphysical and UVB radiative impacts"

_EGUsphere, 2025_

## Referee Comment (RC1)

Gidarakou et al. use in situ and ground-based measurements collected during the BIOSPHERE campaign, in combination with satellite observations and FLEXPART simulations, to investigate the optical and microphysical properties of transported dust and smoke aerosols and their mixtures. These observations are highly relevant for constraining radiatively important intensive optical properties, such as the single-scattering albedo and lidar ratio of dust and smoke originating from wildfires. I find the methodology to be sound and the interpretation of the results convincing. Uncertainties associated with the observations are appropriately documented where necessary. The following comments are minor and are intended to further improve the manuscript. Given the focus and scope of the study, I recommend publication in *Atmospheric Chemistry and Physics* as a "Measurement Reports" paper after the authors have addressed the comments below.

Compare case studies of transport of both dust and smoke and of fresh smoke particles over Athens.

**Comments:**

1. **Line 70:** The abbreviation "EFFIS" appears for the first time here and is not defined.
2. **Line 72:** correct "wildfires" to "wildfire"
3. **Lines 83–86:** This paragraph could be expanded to better convey the significance of the synergistic use of multiple observational platforms. Clarifying the complementary strengths of each sensor would help demonstrate the added value of their combined use. Such synergy also offers an opportunity to assess consistency among sensors and to identify their respective limitations in retrieving the properties of dust and smoke mixtures, which remain challenging to characterize using satellite observations alone. Also, the role of FLEXPART in relation to the various observations can be introduced here. Including these aspects would provide readers with clearer expectations regarding the scope and objectives of the study.
4. **Line 107:** How were the observations from the two independent lidar systems used in this study?
5. **Line 184:** Please clarify whether observations from both depolarization and Raman lidar retrievals were used to retrieve aerosol microphysical properties.
6. Since the manuscript uses multiple platforms with overlapping observations, I strongly recommend the inclusion of a schematic explaining the data collection and processing chain, as it would substantially improve the readability of the manuscript. At its present form, I find it hard to navigate and find appropriate

information in Section 2 regarding any particular property used in the results section.

7. **Lines 235-243**: Please cite the appropriate references for this information, for example an online article.
8. **Lines 278-286**: This is repeated in the next paragraph.
9. **Line 355**: Correct "from18"
10. **Figure 4 and Section 3.3**: Consider adding tick marks to all panels. I also suggest merging the panels into a single figure, as they share the same axes and color bar limits. The merged curtain plot could display timestamps at 6-hour intervals without the minute component (DD–HH), with minor ticks every hour. A second row could be used to show the simultaneous particle depolarization ratio retrievals or lidar ratio retrievals, which would allow for a more accurate assessment of potential aerosol types. In its current form, the association between RCS and aerosol type is not sufficiently clear in the manuscript.
11. **Figure 5, panels c, e and f:** Adjust x axis limits.
12. **Line 390:** Please mention the time between which the profiles are averaged and the reason behind this choice in Figure 5.
13. **Line 410**: Could you also comment on the peak in $A_{b355/532}$ in the upper layer.
14. Please follow comment 10 for Figure 8 as well.
15. As a general comment, I recommend shortening the manuscript. At present, the two case studies are described in great detail separately, which makes direct one-to-one comparison difficult. While a complete restructuring of the Results section may be beyond the scope of this review, I suggest that the authors reduce repetitive text and consolidate information that leads to the same conclusion. For example, both a lower Ångström exponent and a higher depolarization ratio indicate the presence of coarse dust; combining these complementary pieces of evidence instead of mentioning them separately would improve clarity while also shortening the manuscript.